# Evaluating rural sports tourism competitiveness: A framework applied to Shandong province

**Lijun Yin[1], Qian Gu[2], Xueting Gao[1], Pengfei Tai[1], Li Cao[3]\***

**1** School of Sport Science, Qufu Normal University, Jining, China, **2** School of Physical Education, Shandong University, Jinan, China, **3** School of Physical Education, Shandong Normal University, Jinan, China

\* li207856@126.com

## Abstract

Rural sports tourism performs a crucial role in developing the rural sports industry and promoting rural revitalization. We investigate the competitiveness of rural sports tourism in Shandong Province through niche theory, developing a five-tier index system in the context of resource base, industrial development, location conditions, fundamental guarantees, and construction management. A least squares optimization model is used to identify both subjective and objective weights for the indicators. In detail, we evaluate the niche intensity and overlap of 16 cities, suggesting significant disparities in competitiveness. Higher niche overlap correlates with enhanced competition, emphasizing the need for better resource development, diversified economic formats, as well as stronger location conditions, basic guarantees, marketing management, and city-level cooperation. These findings not only offer practical recommendations for enhancing competitiveness, but also provide theoretical guidance for the high-quality development of regional rural sports tourism.

## 1. Introduction

In China, the 20th CPC National Congress demonstrated that developing rural characteristic industries is essential for advancing rural construction [1]. Sport, as a key component of rural industries, performs a crucial role in rural development. Besides, "No. 1 central document" for 2022 emphasized the need to "promote steady and healthy economic and social development, along with the continuous and comprehensive advancement of rural revitalization." Rural sports tourism integrates sports with rural industries, leveraging local cultural, natural, and sports resources to induce both rural revitalization and economic growth [2]. In the context of global development, the relevant concepts are not only applicable to China's national conditions, but also applicable to a significant trend and guiding plan for global rural development. However, rural sports tourism faces significant challenges, due to the lack of an effective evaluation system and development plan. Issues, such as poor tourist

**Data availability statement:** All relevant data are within the manuscript and its Supporting Information files.

**Funding:** The author(s) received no specific funding for this work.

**Competing interests:** The authors have declared that no competing interests exist.

experiences, inadequate regional cultural development, and weak and construction management, impede progress [3]. The progression of rural sports tourism needs substantial improvement. Competitiveness in this sector signifies the ability to market attractions, to provide service guarantees, and to establish supporting facilities, hence generating economic benefits. Analyzing the current status and competitiveness of rural sports tourism and providing targeted countermeasures and recommendations may offer valuable theoretical guidance for sustainable development and ensure the effective implementation of practical initiatives. These approaches are vital for boosting the high-quality development of rural sports tourism and executing the rural revitalization strategies.

## 2. Literature review

The research on rural sports tourism competitiveness has been centered on the academic field, which is based on the research on tourism destination competitiveness. There is an overview of the research on tourism destination competitiveness, in detail, comprising definitions, analysis of influencing factors, and evaluation methods [4–7]. In terms of definition, scholars have attempted to define the concept of tourism destination competitiveness from various perspectives, emphasizing its ability to attract tourists and investments on a global scale. The assessment of influencing factors has deeply explored the impact of natural landscape, cultural history, tourism facilities, and transportation convenience on the competitiveness of tourist destinations [8,9]. The research on evaluation methods involves a variety of quantitative and qualitative tools, such as AHP-entropy weight method and AHP analysis [10,11], etc. Overall, the above methods provide technical support for the evaluation of tourism destination competitiveness and lay a theoretical research foundation for the evaluation of rural sports tourism competitiveness.

In addition, a comprehensive search on the theme of "rural sports tourism competitiveness evaluation" indicates that the relevant research topics are mainly concentrated on rural tourism competitiveness and sports tourism competitiveness. There are relatively few targeted studies on the evaluation of rural sports tourism competitiveness, and they are concentrated on qualitative research on rural sports tourism. The research content is mainly on the rural sports consumption market [12,13],Construction of rural sports tourism communities [14,15],Cultural adhesion of rural sports events [16,17]、 People-land (resident) interaction in rural sports tourism [18,19],Sustainable development of rural green (ecological) sports and its poverty alleviation effect [20,21]. Therefore, to scientifically carry out the research on the evaluation of rural sports tourism competitiveness, we can refer to the research on the construction of the evaluation index system of sports tourism and rural tourism competitiveness in the previous dimension, and carry out in-depth research in combination with the theory formed by its related methods and content, so as to enrich the theoretical research framework.

Research methods on sports tourism competitiveness are mostly centered on quantitative analysis, and the main research content is to establish models to identify influencing factors, using research methods to evaluate competitiveness. At the

level of influencing factors, the constructed model is utilized to identify destination supporting factors, restrictive factors, and enhancing factors [22,23], and key factors of consumer participation [24], participation motivation [25]. Thereby, this facilitates the clarification of the factors that affect the development of competitiveness of sports tourism destinations. The competitiveness level is evaluated through factor analysis [26], focus groups [27], model construction [28], and additional methods are used to scientifically evaluate the competitiveness of sports tourism destinations. The research methods around the evaluation of rural tourism competitiveness also focus on quantitative analysis, mainly using quantitative evaluation methods such as building models and indicator systems to analyze regional rural tourism competitiveness. In terms of model construction, the integrated multi-source data and Michael Porter diamond model are used to evaluate the competitiveness of rural tourism in Henan Province [29]. Besides, the World Economic Forum's Travel and Tourism Competitiveness Report (TTCR) model is used to compare and analyze the competitiveness of rural tourism in Indonesia [30]. Moreover, a comparative assessment of Poland's competitiveness with selected European countries using a composite tourism competitiveness index and panel regression to measure rural tourism competitiveness [31]. In addition, inherited resources, created resources, supporting factors, destination management [32], demand conditions and situational conditions [33] are used to measure and compare the rural tourism competitiveness of different countries and regions. In summary, the academic community has delved into the influencing factors, competitiveness evaluation of sports and rural tourism destinations, and the status quo of rural sports tourism, highlighting their significance in regional development. In research methods, scholars apply interdisciplinary approaches, enriching theory, yet few have designed a competitiveness evaluation index system for rural sports tourism. This study focuses on Shandong's rural sports tourism competitiveness. It constructs an evaluation index system via the least - squares optimization model, applies niche theory to assess competitiveness in 16 cities, and uses the results to suggest strategies, offering reference for rural sports tourism development in Shandong and elsewhere.

## 3. Materials and methods

### 3.1 Theoretical basis

Niche theory is an ecological framework used to assess the relationship between species and their external environment. It quantitatively reflects the spatial position and functional role of species within their ecosystems [34]. Within tourism research, the niche theory has evolved from its original concept [35] and has been widely used in various related fields such as tourism industry competitiveness evaluation [36], urban tourism competition [37], tourism planning and development [38], regional tourism cooperation [39], tourist attraction evaluation [40], and sustainable tourism development [41]. It can objectively clarify the vertical and horizontal relationships, logical connections and changing laws of various elements in the tourism ecosystem through quantitative relationships [42]. In summary, it is theoretically feasible to transfer the ecological niche theory to rural sports tourism. Based on the theoretical analysis of ecological niche, in the context of rural sports tourism, which is a complex ecosystem involving factors like industry, resources, operations, management, and market, each city acts as an ecological unit that either competes with or supports others. This study defines the "city-level rural sports tourism competitiveness niche" as a metric for assessing a city's development status, spatial position, and functional value in the tourism ecosystem. A comprehensive evaluation of this niche can reveal the status and function of rural sports tourism from multiple dimensions and clarify the ecological value of each factor within the system.

 **3.1.1 Niche strength.** Niche strength refers to the ability of species to compete for shared resources [43], including the resources and environmental factors necessary for survival and development. In urban-rural sports tourism, niche strength reflects a city's capacity to acquire, process, develop, and control resources within the rural sports tourism ecosystem, considering environmental conditions and spatial-temporal factors. This strength provides an ecological advantage in real-world competition. Niche intensity can effectively measure the competitiveness of cities in rural sports tourism and reflect the differences in competitiveness and the competitive landscape.

**3.1.2 Niche overlap.** Niche overlap refers to the competition between two species for the same resources in the same space. The greater the niche overlap, the more intense the competition for those resources, and vice versa [44]. In niche overlap theory, this competition stems from similarities in resource use, spatial-temporal locations, and market conditions [45,46]. In this study, niche overlap in urban-rural sports tourism is defined as the competition for tourism resources among two or more urban areas as they respond to societal, economic, and environmental factors. Niche overlap can effectively measure the competitive relationship between cities and suggest the potential for regional tourism cooperation, in detail, in terms of resource similarity and market positioning, and provide theoretical support for resource sharing and coordinated development among cities.

## 3.2 Methods

**3.2.1 Indicator construction.** Rural sports tourism is a new form of business that relies on rural sports resources and meets the needs of tourism. Evaluating the competitiveness of urban rural sports tourism is helpful to promote the development of rural sports industry, and building an objective and reasonable evaluation system is the key to measuring competitiveness [47]. Based on the analysis of multidimensional hypervolume niche theory, the ecological niche of urban rural sports tourism determines its position in the tourism destination ecosystem, and its size is jointly determined by tourism resources, market position, social economy, ecological environment and other factors [48]. This study draws on domestic and foreign research on the evaluation of competitiveness factors, including resource endowment in rural tourism competitiveness [49], industrial agglomeration and business innovation in sports tourism [24], location conditions in tourism destination competitiveness [8], infrastructure and public services in rural tourism [29], and marketing and market supervision in tourism destination management [28]. At the same time, combined with the international urban rural sports tourism practice, following the principles of scientificity, comprehensiveness and operability, a three-layer urban rural sports tourism competitiveness evaluation index system was constructed.

Among them, it should be pointed out that in the specific calculation process of the evaluation index, some indicators are reflected by weighted values, such as "rural sports tourism resource popularity" and "rural sports tourism demonstration creation unit". Since popularity is difficult to quantify directly, it is necessary to convert it into comparable values. On the one hand, according to the national standard "Classification, investigation and evaluation of tourism resources" (GB/T 18972–2017), the "popularity and influence" factor in the tourism resource evaluation system is measured by national and provincial resources. Among them, the score of national resources is usually between 5–7 points, and the score of provincial resources is between 3–4 points, that is, the score of national resources is about 1.5–2 times that of provincial resources [50]. At the same time, referring to the rural tourism resource evaluation research of Wang & Zhu (2019) [49] and L Mo (2017) [51], the weight difference between national and provincial resources is set to 2. On the other hand, the resource diversity theory emphasizes the richness and diversity of resources, and is specifically applied to the evaluation of tourism resource attractiveness [52], tourism resource popularity [53] and other studies, which provides a useful reference for the selection of resource level evaluation popularity in this study. In summary, the calculation method of the above two indicators in this study is determined to be "the number of national resources × 10 + the number of provincial resources × 8".

The system consists of three layers: the system layer reflects overall competitiveness, the dimension layer includes five aspects—resource base, industrial development, location conditions, basic guarantees, and construction management—and the indicator layer features 13 specific indicators (see Fig 1 for details).

**3.2.2 Evaluation index description.**

1. Resource Basis Dimension: This dimension assesses the quantity and quality of tourism resources that support rural sports tourism development. It includes three indicators: the quality of rural sports tourism resources (X11), the number of rural sports tourism resources (X12), and the popularity of rural sports tourism resources (X13). The quality of

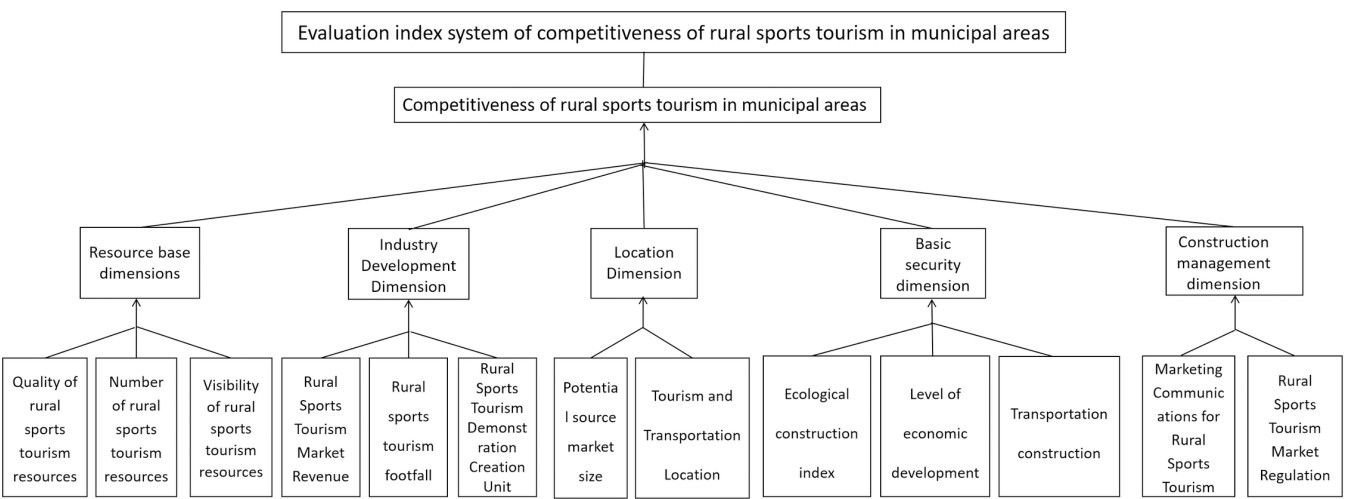

**Fig 1. Evaluation index system of urban-rural sports tourism competitiveness.**

rural sports tourism resources (X11) refers to natural landscapes such as mountains, water bodies, forests, fields, lakes, grasslands, and sand, as well as cultural landscapes characterized by village settlements, folk customs, and farming activities, all of which contribute to rural sports tourism within a city. This indicator is measured by counting the types of available resources. The number of rural sports tourism resources (X12) refers to the total quantity of such resources within a city, measured in units. The popularity of rural sports tourism resources (X13) reflects the level of development and promotion of rural sports tourism, calculated as "number of national rural sports tourism resources × 10 + number of provincial rural sports tourism resources×8."

2. Industrial Development Dimension: This dimension reflects the economic impact of the rural sports tourism industry. It includes rural sports tourism passenger flow (X21), rural sports tourism market income (X22), and rural sports tourism demonstration units (X23). Rural sports tourism passenger flow (X21) refers to the total number of tourists participating in rural sports tourism within a city, measured in tens of thousands of people per year. Rural sports tourism market income (X22) represents the economic revenue generated by rural sports tourism, measured in hundreds of millions of yuan per year. Rural sports tourism demonstration units (X23) indicate the level of development and construction of high-quality demonstration units in the city, including national and provincial leisure agriculture sites, rural tourism demonstration points, and sports-related scenic spots. This indicator is calculated as "number of national demonstration points×10 + number of provincial demonstration points×8."

3. Location Condition Dimension: This dimension reflects the market size and transportation accessibility of rural sports tourism within a city. It includes the potential source market size (X31) and tourism transportation location (X32). The potential source market size (X31) is determined by calculating the urban active population within a 1-hour transportation radius from the city center. The tourism transportation location (X32) is assessed based on the average distance between high-level rural sports tourism demonstration units within the city and both the city center and surrounding areas.

4. Basic Guarantee Dimension: This dimension reflects the external conditions supporting the development of rural sports tourism in a city. It includes three indicators: economic development level (X41), ecological construction index (X42), and transportation infrastructure (X43). The economic development level (X41) is measured by the city's annual GDP, expressed in hundreds of millions of yuan. The ecological construction index (X42) is evaluated based

on the comprehensive forest coverage rate and air quality. Transportation infrastructure (X43) is assessed by the density of graded highways within the city, measured in kilometers per square kilometer (km/km²).

5. Construction and Management Dimension: This dimension is evaluated through two indicators: rural sports tourism marketing and communication (X51) and rural sports tourism market supervision (X52). These indicators reflect the level of market supervision, marketing efforts, promotional activities, and investment in rural sports tourism by municipal government departments. Rural sports tourism marketing and communication (X51) is measured by the number of large-scale exhibitions and sales events held annually in the municipality. Rural sports tourism market supervision (X52) is assessed by the number of relevant complaint cases handled annually within the municipality.

### 3.2.3 Subjective and objective comprehensive weighting based on least squares optimization

model. Determining the weight of evaluation indicators is crucial for developing an evaluation indicator system. Weighting methods are typically classified as either subjective or objective. Subjective weighting involves expert opinions, which, though scientifically valid, rely on personal judgment and may obscure objective data. Objective weighting methods use data but may overlook qualitative factors. Each approach has its strengths and limitations. Therefore, this study employs the least squares method to combine both subjective and objective techniques, improving the scientific accuracy of the evaluation indicator weights.

1. Determining subjective weights: AHP

The Analytic Hierarchy Process (AHP) is a decision-making approach that breaks decision factors into hierarchical levels for analysis. It efficiently evaluates relationships among multiple factors within a system and is valued for its practicality, simplicity, and structured organization. As a subjective weighting method, The specific operation steps of the AHP hierarchical method are as follows [54]:

First, construct the hierarchical structure of the indicator system. According to the AHP hierarchical analysis method, the rural sports tourism competitiveness evaluation index system is divided into the system layer, dimension layer and index layer.

Secondly, construct the judgment matrix. This article invited a total of 10 scholars and practitioners with more than 10 years of research experience in rural sports tourism, tourism management or related fields. These experts came from different cities in Shandong Province to ensure regional representativeness and the stability and reliability of the results. Usually, experts need to compare the indicators of the same level in pairs and make scoring judgments based on the relative significance [55]. The specific interpretation of each score is shown in Table 1. The significance of indicators at different levels and between different levels is judged respectively, and the weight of each indicator is calculated using the judgment matrix formed by expert scoring.

Thirdly, conduct a consistency test. Due to the inconsistency of the number of levels, some levels may have too many indicators, which will interfere with the scientific scoring of experts and lead to the inconsistency of the judgment logic before and after the indicators. Therefore, it is necessary to conduct a consistency test on the constructed judgment matrix to eliminate the logical confusion caused by subjective scoring and obtain a weight with higher credibility [56] (Table 2). The consistency test calculation formula is:

$$CR = \frac{CI}{RI}$$

$CI$ is the consistency index, and its calculation formula is: $CI$ = (largest characteristic root - n) (n-1), $RI$ is the random consistency index, and $CR$ is the consistency ratio. Different orders correspond to different RI values. When the order is greater than 2, it constitutes a random consistency ratio, and only then is a consistency test required. When $CR$ is less than 0.1, it proves that the judgment matrix can pass the consistency test and the subjective scoring of the experts is

**Table 1. The specific meaning of each score.**

| Assignment | significance level | Remark |
|---|---|---|
| 1 | Both indicators are equally significant | 2, 4, 6, 8, 1/2, 1/4, 1/6, 1/8 indicate that their relative significance is between the adjacent values of 1, 3, 5, 7, 1/3, 1/5, 1/7 |
| 3 | The former is slightly more significant than the latter | |
| 5 | The former is obviously more significant than the latter | |
| 7 | The former is strongly more significant than the latter | |
| 9 | The former is more significant than the latter | |
| 10 | The former is slightly less significant than the latter | |
| 11 | The former is obviously less significant than the latter | |
| 12 | The former is strongly less significant than the latter | |
| 13 | The former is extremely less significant than the latter | |

**Table 2. Random consistency RI value [57].**

| Degree | 1 | 2 | 3 | 4 | 5 | 6 | 7 | 8 |
|---|---|---|---|---|---|---|---|---|
| RI value | 0 | 0 | 0.52 | 0.89 | 1.12 | 1.26 | 1.36 | 1.41 |

reasonable. When $CR$ is greater than 0.1, it proves that the consistency of the matrix is poor, and the matrix needs to be adjusted until the consistency is qualified.

2. Determining objective weights: Information entropy method.

The information entropy method quantifies the content of information in a system to help optimize targets or select parameters. It assigns objective weights based on the variability of indicators, reflecting the information they convey. In an evaluation system, lower information entropy means a higher weight for the indicator, while higher entropy indicates a lower weight [58].

The information entropy method can be applied to determine objective weights through the following three steps:

First, the quantitative values of each indicator are determined, forming a weight decision matrix that is then standardized. Assuming there are o samples to be evaluated and p evaluation indicators, let $x_{ij}$ (i = 1,2,3…, o; j = 1, 2, …n) represent the quantitative value of sample i on indicator j. This allows the formation of the weight decision matrix $X$.

$$X = \begin{bmatrix} x_{11} & x_{12} & \dots & x_{13} \\ x_{21} & x_{22} & \dots & x_{23} \\ \dots & \dots & \dots & \dots \\ x_{m1} & x_{m2} & \dots & x_{mn} \end{bmatrix}$$

To standardize the evaluation indicators and eliminate incommensurability between dimensional units, the index data is normalized. This study employs the range standardization method, with the formula as follows:

$$\text{For positive indicators: } x'_{ij} = \frac{(x_i - x_{\min})}{(x_{\max} - x_{\min})} \tag{1}$$

$$\textit{For } \text{negative indicators: } x'_{ij} = \frac{(x_{\max} - x_i)}{(x_{\max} - x_{\min})} \tag{2}$$

In this formula, $y_{ij}$ represents the standardized weight decision matrix, where $x_{\max}$ and $x_{\min}$ denote the quantized values of the maximum and minimum for different samples under the same indicator.

Secondly, the information entropy for each evaluation index is determined. Let $F_j$ represent the information entropy of the jth index. The calculation formula is as follows:

$$F_j = -\frac{1}{\ln o} \sum_{i=1}^{o} h_{ij} \ln h_{ij}$$

(3)

Here, $h_{ij} = \frac{y_{ij}}{\sum_{i=1}^{n} y_{ij}}$, with the stipulation that when $y_{ij}=0, h_{ij} \cdot \ln h_{ij}=0$.

Finally, the objective weight value of each indicator, denoted as $z_j$, is calculated based on the information entropy value $F_j$. This value represents the entropy of the jth indicator, and the calculation formula is as follows:

$$z_j = (1 - F_j) / \sum_{j=1}^{n} (1 - F_j)$$

(4)

3. Least squares optimization model to determine comprehensive weights

It is assumed that the subjective weight of the evaluation index is determined through hierarchical analysis:

$$K = [k_1, k_2, ..., k_n]^T$$

The objective weight of the evaluation index is determined using the information entropy method:

$$L = [l_1, l_2, ..., l_n]^T$$

To integrate the information reflected by both subjective and objective factors and ensure their alignment, it is essential to minimize the deviation between the subjective and objective weights in determining the final comprehensive weight. After comprehensive consideration, to balance the influence of subjective weight (expert opinion) and objective weight (data-driven) and avoid over-reliance on a single source of weight, this study chose $\beta = 0.5$, that is, the subjective weight and objective weight each accounted for 50%. This study employs the least squares method to reconcile the subjective and objective weights, constructing a least squares model:

$$\begin{cases} minF(W) = \sum_{i=1}^{o} \sum_{j=1}^{p} \{[(k_j - w_j)y_{ij}]^2 + [(z_j - w_j)y_{ij}]^2\} \\ st. \sum_{j=1}^{p} w_j = 1 \\ w_j \geq 0, j = 1, 2, ..., n \end{cases}$$

(5)

In this formula, wj represents the comprehensive weight, kj denotes the subjective weight, zj is the objective weight, and yij refers to the standardized decision matrix.

4. Significance analysis and robustness testing

To verify the significance and robustness of the estimates obtained by the least squares method [59], statistical significance test: residual analysis, goodness of fit ($R^2$) test and F test were implemented to evaluate the overall fit effect and statistics of the model Significance; perform weight sensitivity analysis, data perturbation test and subsample analysis to explore the impact of key parameter changes and data changes on model output and evaluate the robustness of the results.

(1) Significance analysis

The purpose of significance analysis is to test whether the fitting effect of the least squares model is significant, that is, whether the model can effectively explain the variation of the data.

 

First, perform residual analysis. The residual is the difference between the actual value and the model predicted value. By analyzing the residual, the fitting effect of the model can be judged. The calculation formula is:

$$\text{Residual} = w_j - (\beta \cdot + (1-\beta) \cdot v_j)$$

Among them, $w_j$ is the comprehensive weight, $u_j$ is the subjective weight, $v_j$ is the objective weight, and $\beta$=0.5

Secondly, the goodness of fit (R²) test is performed. The goodness of fit (R²) is used to measure the model's ability to explain the data. The closer R² is to 1, the better the model fits. The calculation formula is:

$$R^2 = 1 - \frac{\sum_{j=1}^{n} (w_j - \hat{w}_j)^2}{(w_j - \overline{w})^2}$$

Among them, $\hat{w}_j = 0.5 \cdot u_j + 0.5 \cdot v_j, \overline{w}$ is the mean of the comprehensive weight.

Next, an F test is performed to test the overall significance of the model, that is, whether the model is significantly better than a simple model that only uses the mean for prediction, and the F statistic is calculated:

$$F = \frac{\text{Explained Variance}}{\text{Residual variance}} = \frac{\sum_{j=1}^{n} \frac{(\hat{w}_j - \overline{w})^2}{k}}{\sum_{j=1}^{n} \frac{(w_j - \hat{w}_j)^2}{(n-k-1)}}$$

Where n=1, k=1.

(2) Robustness testing

Robustness testing aims to verify the stability of the model under different conditions, that is, to ensure the reliability of the research conclusions.

First, a weight sensitivity analysis is conducted. The weight sensitivity analysis tests the changes in the comprehensive weight by adjusting the combined coefficient of the subjective weight and the objective weight. This paper sets β = 0.3, β = 0.5, and β = 0.7 to calculate the corresponding comprehensive weights and compare their changes. The calculation formula is:

$$\hat{w}_j = \beta \cdot u_j + (1-\beta) \cdot v_j$$

Secondly, a data perturbation test is conducted. The data perturbation test adds random noise (±5% perturbation) to the subjective weight and objective weight, recalculates the comprehensive weight, and tests the sensitivity of the model to data perturbations. For each indicator's subjective weight $u_j$ and objective weight $v_j$, add ±5% random noise to generate perturbed weights $u_j'$ and $v_j'$, use the perturbed weights to recalculate the comprehensive weight $w_j'$, and compare it with the original comprehensive weight $w_j$.

Next, we conduct sub-sample analysis. By dividing the data into several sub-samples (such as grouping by city area), we calculate the comprehensive weights for each sub-sample and test the stability of the model under different sub-samples.

### 3.2.4 Calculation of niche intensity and overlap.

1. By calculating the rural sports tourism niche intensity of each city, we can objectively reflect the relative position of the city in resource competition. Specifically, by using weighted standardized indicator data, the niche intensity calculation can comprehensively reflect the competitiveness of the city in five dimensions: resource base, industrial development,

location conditions, basic guarantees, and construction management. The niche intensity calculation formula is as follows:

$$S_i = \sum_{j=1}^{K} x'_{ij} w_j$$

(6)

Where Si denotes the rural sports tourism niche intensity for the ith municipality, $x'_{ij}$ represents the standardized indicator data, and $w_j$ corresponds to the weight of the jth indicator.

2. Due to the different competitive pressures of cities at different levels, the niche overlap they present is also different, that is, $B_{op} \neq B_{po}$. Usually, the MacArthur model is often used to calculate the niche overlap. The model used in this study is different from the standard MacArthur model. It is mainly used to analyze the competitiveness of rural sports tourism. By calculating the score ratio of city o and city p in various indicators, the competitive relationship between the two in terms of resources, industry and location is measured. The larger the value, the more intense the competition between city o and city p. The formula for calculating the niche overlap is as follows:

$$B_{op} = \frac{\sum_{j=1}^{R} P_{pj} P_{oj}}{\sum_{j=1}^{R} P_{oj}^2}$$

(7)

Where $B_{op}$ represents the ecological niche overlap value of rural tourism in city o to rural tourism in city p, and $P_{pj}$ and $P_{oj}$ represent the ratio of index j in the evaluation index score of city p and city o, respectively.

## 3.3 Study area and data sources

**3.3.1 Overview of the study area.** As a major economic province on the east coast of China, Shandong Province has demonstrated remarkable development vitality and potential in the field of sports tourism in recent years. Thanks to the government's active policy guidance and the implementation of the strategy of using sports to help rural revitalization, Shandong Province has made remarkable achievements in the construction of sports tourism demonstration bases and the integrated development of sports and tourism with its superior geographical location and rich resource types. Shandong Province has 3,500 large-scale rural tourism sites and 33 national demonstration sites, ranking first in the country. During the "14th Five-Year Plan," the province plans to establish 10 national sports tourism demonstration bases and recognize 30 provincial sports tourism bases, providing a rich empirical basis for academic research on sports tourism. These efforts suggest strong development potential for rural sports tourism.

This study analyzes 16 cities in Shandong Province. Using the evaluation index system and the least squares method, both subjective and objective weights are determined. The competitiveness of rural sports tourism in each city is calculated based on its development, assessing strengths and overlaps. An overall evaluation of rural sports tourism competitiveness in the province is then completed.

**3.3.2 Data sources.** The data for evaluating rural sports tourism development indicators come primarily from the 2023 statistical yearbooks of 16 cities in Shandong Province. Additional information is sourced from public documents and reports from the Shandong Provincial Department of Culture and Tourism, the Shandongovincial Department of Natural Resources, the Shandong Provincial Bureau of Statistics, and municipal culture and tourism bureaus. Some unpublished data are obtained through direct communication with relevant departments via phone and email.

# 4. Results

## 4.1 Niche strength of rural sports tourism competitiveness

To enhance the accuracy and reliability of subjective evaluations, reduce errors, and limit the number of indicators requiring individual expert assessments, experts in relevant fields compare indicators both within the same dimension and across different dimensions. The hierarchical analysis method is used to derive subjective weights from a judgment matrix that meets consistency tests. The final subjective weight vector for each indicator is obtained by multiplying these derived weights. The original data matrix X is standardized using formulas (1) and (2) to produce a standardized decision matrix. The objective weight for each indicator is calculated with the entropy weight method, as described in formulas (3) and (4). These values are then input into the least squares calculation formula (5), and the comprehensive weight of each indicator is determined using MATLAB software, as shown in Table 3.

The results of significance analysis showed that the residuals followed a normal distribution (Shapiro-Wilk test p = 0.996), indicating that the model had a good fit. The goodness of fit is high ($R^2 = 0.999$), indicating that the model can explain the variation in the data well. The model is significant (F-statistic = 12960.0), indicating that the model is significantly better than the simple model that only uses the mean for prediction. Therefore, the least squares model proposed in this paper is highly scientific and reliable, and its comprehensive weights can provide reliable support for evaluating rural sports tourism competitiveness. The robustness test results show that the least squares model proposed in this study shows good stability under different conditions. Whether it is weight sensitivity analysis, data perturbation test, or sub-sample analysis, the change range of the comprehensive weight is small, indicating that the model is insensitive to parameter changes, data perturbation, and sub-sample grouping. Therefore, the comprehensive weights determined in this study are highly reliable and robust.

## 4.2 Competitiveness niche intensity of rural sports tourism

### 4.2.1 Overall analysis of the ecological niche strength of rural sports tourism competitiveness.
After designing the index system with ecological niche theory and calculating comprehensive weights using the least squares method, standardize the decision matrix according to formula (6). Compute the competitiveness of rural sports tourism in Shandong Province's cities (see Fig 2). The city rankings are: Qingdao > Jinan > Yantai > Jining > Linyi > Weifang > Weihai > Zibo > Tai'an > Rizhao > Zaozhuang > Heze > Liaocheng > Dezhou > Binzhou > Dongying.

Table 3. Subjective, objective, and comprehensive weights of evaluation indicators.

| Evaluation indicators | Subjective weight $u_j$ | Objective weight $v_j$ | Comprehensive weight $w_j$ |
|---|---|---|---|
| X11 | 0.0283 | 0.0523 | 0.0403 |
| X12 | 0.0869 | 0.0882 | 0.0875 |
| X13 | 0.1777 | 0.0440 | 0.1109 |
| X21 | 0.0331 | 0.0618 | 0.0475 |
| X22 | 0.1045 | 0.1260 | 0.1153 |
| X23 | 0.2085 | 0.0400 | 0.1243 |
| X31 | 0.1443 | 0.0220 | 0.0831 |
| X32 | 0.0525 | 0.0150 | 0.0337 |
| X41 | 0.0178 | 0.1517 | 0.0848 |
| X42 | 0.0591 | 0.0389 | 0.0490 |
| X43 | 0.0493 | 0.1095 | 0.0794 |
| X51 | 0.0230 | 0.2172 | 0.1201 |
| X52 | 0.0149 | 0.0334 | 0.0242 |

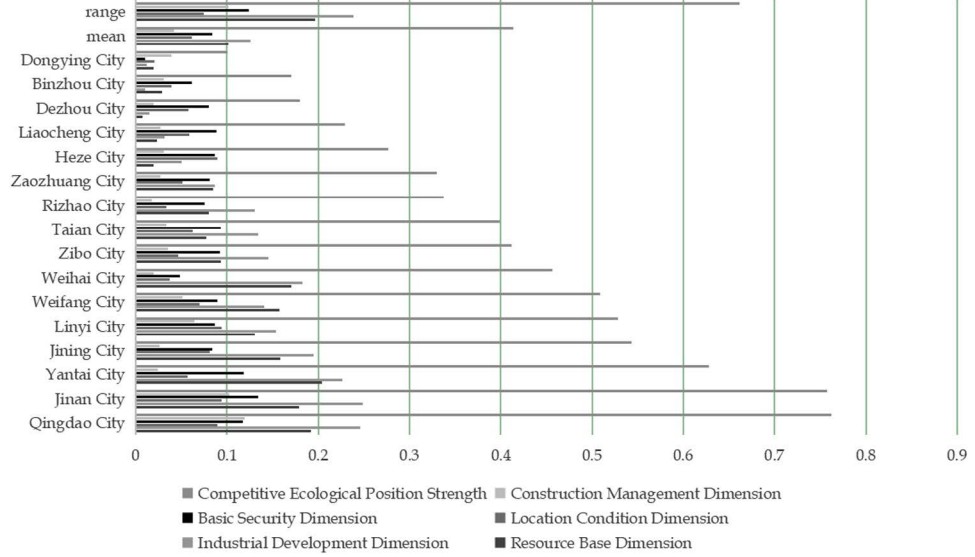

**Fig 2. Niche strength of rural sports tourism competitiveness in various cities in Shandong Province.**

Based on the ranking analysis, Qingdao ranks first due to its well-balanced rural sports tourism resources. Notably, 13 villages in Qingdao are part of the fourth batch of scenic villages in Shandong Province. The city hosts 4 listed companies and 2 national-level specialized and new enterprises, providing strong industrial conditions that support rural sports tourism. In 2023, the per capita disposable income of rural residents in Qingdao reached 29,736 yuan, reflecting high market openness. The "Opinions on Building a Shandong Model Pilot Zone for Rural Revitalization and Accelerating Agricultural and Rural Modernization" emphasizes improving public infrastructure in rural areas. Additionally, the Laoshan 100km International Mountain Cross-Country Challenge is repeatedly recognized as a top Chinese sports tourism destination, known for its well-organized management. The "Qingdao Rural Infrastructure Network Construction Action Plan" aims to achieve a rural road rate exceeding 80%, highlighting significant locational advantages.

Jinan ranks second, excelling in industrial development, location conditions, and basic support. These factors align with the standards of a provincial capital city, featuring high public service levels, strong economic growth, comprehensive infrastructure, and a large population. Jinan ranks 15th in the "New Era China City Social Development Index and Top 100 List (2023)" with an A+ rating, making it the highest-ranked city in Shandong. By 2023, Jinan plans 20 rural tourism routes with various themes.

Yantai ranks third, with the highest score in the resource base dimension, indicating its rich and diverse rural sports tourism resources. During the Rural Cultural Tourism Festival, Yantai introduces three major themes and six route products focused on ice and snow, folk customs, and tourism. The city also organizes 20 key municipal activities and 70 district and city activities.

Cities like Jining, Linyi, Weifang, Weihai, Zibo, Tai'an, Rizhao, Zaozhuang, and Heze rank in the middle, showing balanced competitiveness dimensions. In contrast, Liaocheng, Dezhou, Binzhou, and Dongying rank lower due to factors like administrative area size, economic development level, resource base, and industrial development level. For example, in the 2023 GDP ranking of cities in Shandong Province, these cities rank 14th, 10th, 13th, and 9th, respectively, indicating a middle-to-lower level. In terms of administrative area size, these cities rank 10th, 7th, 9th, and 11th, respectively. Overall, these cities have relatively weak developmental foundations.

A comprehensive analysis of niche intensity shows significant differences in rural sports tourism competitiveness across cities in Shandong Province, especially in resource base and industrial development. Resource advantages attract

tourists, while industrial development affects the quality of sports services. Notable differences also exist in basic guarantee and construction management, which impact the efficiency of rural sports tourism and influence tourists' experiences and decisions to return. The location condition dimension shows the least variation, indicating its minimal impact on competitiveness, though it does enhance tourists' access and experience. Cities with higher economic development generally have stronger industry, transportation, and infrastructure, while cities with slower growth face constraints that limit their development. Cities with strong ecological advantages tend to excel in niche competitiveness intensity.

**4.2.2 Analysis of the ecological niche intensity dimension of rural sports tourism competitiveness.** By comprehensively comparing the rankings and averages of each dimension of niche intensity across the cities, several observations can be made:

1. In cities where the resource base dimension scores exceed the dimension mean, there are more rural sports tourism resources in quantity, quality and popularity, which is consistent with the goal of resource protection and development measures proposed in the "Shandong Rural Tourism Development Plan (2021-2025)". For example, in cities with coastal characteristics, Qingdao's 21 marine ranches under construction have been rated as "national demonstration areas", Yantai's two resorts have been selected as national tourist resorts, and Weihai's annual leisure fishery tourists have reached more than 7 million; in cities with traditional and red cultural characteristics, Jining has two world cultural heritage sites and national historical and cultural cities, and Linyi has 5 national red tourism classic scenic spots; Weifang, which has intangible cultural heritage characteristics, has two intangible cultural heritage characteristic tourist routes selected into the provincial list, while other cities below the average have low resource abundance, quantity and taste, fewer city-level characteristic attractions, and resource development and utilization need to be improved.

2. In the cities where the scores of industrial development dimension exceed the average, the rural sports tourism industry has more receptions and operating income, more complete demonstration sites, broader industrial development prospects, and more diverse business formats, showing strong economic effects and industrial development potential, mainly manifested in the industrial upgrading and economic structure optimization of some cities. For example, Jinan has promoted the construction of six rural tourism clusters with sports characteristics, such as Qilu No. 8 Fengqing Road and Wucaishan Village; Qingdao has carried out rural sports carnivals with competitive activities in all agricultural districts and cities in the city, and organized 7 events and 15 competitions, while other cities below the average have weak industrial development, lack of reception, operating income and demonstration sites, fewer business formats, and have not yet formed economic effects.

3. In cities where the location condition dimension exceeds the average level, rural sports tourism has good development prospects and convenient transportation location, and has superior geographical location and transportation conditions, which makes it easier to attract tourists. This is mainly reflected in the optimization of the spatial layout of rural sports tourism in some cities and the enhancement of location advantages. For example, by the end of 2021, the total mileage of rural roads in Qingdao City reached 12,223 kilometers, of which village roads accounted for 71%; in 2023, the mileage of rural roads in Linyi City reached 30,000 kilometers, and 3 national demonstration counties of "Four Good Rural Roads" were created, while other cities below the average level lacked location conditions, had weaker attractiveness, and inconvenient transportation, resulting in a small scale of potential tourists.

4. The cities with higher scores in the basic guarantee dimension have a higher level of economic development, better ecological environment quality, and more complete comprehensive guarantee construction such as public infrastructure construction, which can meet the basic rural sports tourism needs of tourists, thereby enhancing the overall attractiveness of rural sports tourism. The cities with higher scores in the basic guarantee dimension have good infrastructure and public services to provide tourists with a better tourism experience. For example, Zibo's rural roads will be newly built and

renovated for more than 1,000 kilometers in 2021, its GDP ranks 7th in the province, and its water environment ranks first in the province. Other cities below the average level have a lower overall level due to weak infrastructure.

5. In cities where the construction and management dimension exceeds the dimension mean, the marketing, publicity, supervision and management of rural sports tourism are relatively strong, the development is relatively good, the product media publicity and sports tourism service capabilities are relatively strong, and tourists' feedback can be handled more properly. Their effective marketing and management strategies are helpful to enhance the brand influence and market competitiveness of rural sports tourism. For example, at the 2023 Qingdao Rural Sports Carnival, a "New Farmer Sports +" market was set up at the opening ceremony to sell sports and fitness products and organize leisure sports activities; Jinan invests more than 30 million yuan each year to create characteristic rural tourism routes, while other cities below the average need to improve their publicity, marketing and service levels.

**4.2.3 Hierarchical analysis of niche intensity of rural sports tourism competitiveness.** Using Q-type cluster analysis in SPSS26.0 to measure the similarity between cities based on distance (as illustrated in Fig 3), we can establish a hierarchy of rural sports tourism competitiveness across cities in Shandong Province, classifying them into different levels.

The cities in Shandong Province are divided into six levels based on the competitiveness niche strength of various dimensions: the first tier includes Qingdao and Jinan; the second tier consists of Jining, Linyi, Weifang, and Yantai; the third tier comprises Zibo, Tai'an, and Weihai; the fourth tier includes Rizhao and Zaozhuang; the fifth tier consists of Heze and Liaocheng; and the sixth tier includes Dezhou, Binzhou, and Dongying. Among them, the first layer belongs to the most competitive urban areas. Qingdao and Jinan have outstanding performance in five dimensions: resource base, industrial development, location conditions, basic guarantees and construction management. The niche intensity is

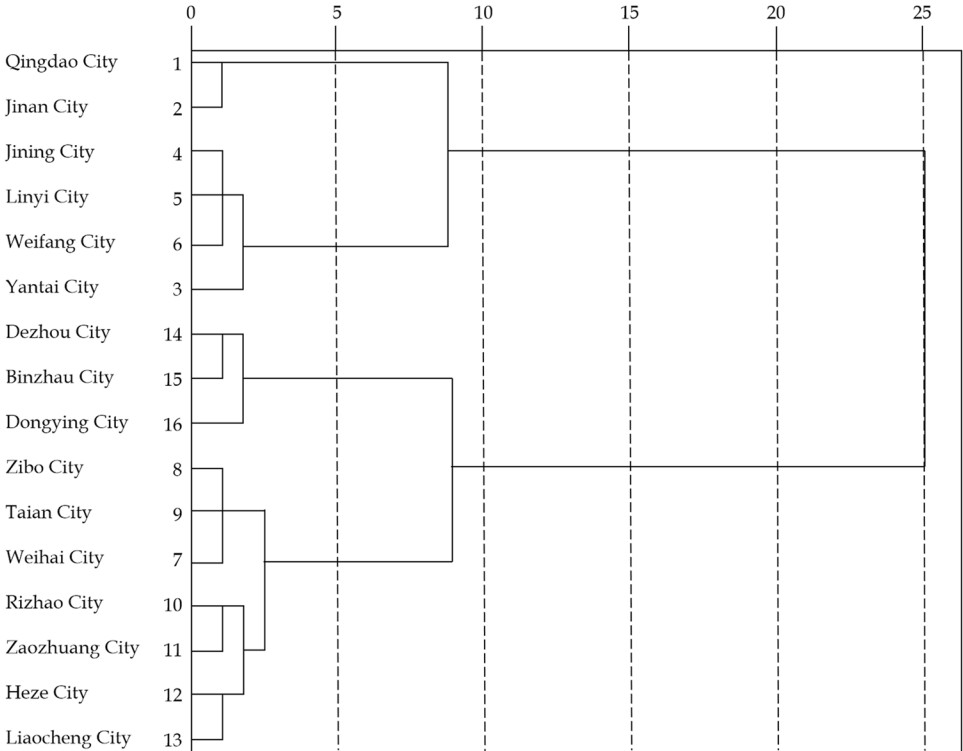

**Fig 3. Cluster pedigree of ecological niche intensity of rural sports tourism competitiveness in Shandong Province.**

significantly higher than that of other urban areas, so they are divided into the first layer. Qingdao and Jinan have performed outstandingly in the five dimensions of resource base, industrial development, location conditions, basic guarantees, and construction management. Their niche intensity is significantly higher than that of other cities, so they are classified as the first level. Although geographically distinct, both cities have made substantial investments in economic development, resources, markets, industry, and management, building a strong foundation for their competitive strength. Their comparable levels of development have also fostered significant competition between them.

In contrast, the sixth-tier cities—Dezhou, Binzhou, and Dongying—exhibit the weakest competitiveness, constrained by underdeveloped economies, limited resource bases, and slower growth in management and service sectors, leading to lower overall niche strength and competitiveness, so they were classified as the sixth level. The cities in the second, third, fourth, and fifth tiers display gradually decreasing competitiveness compared to the first tier. However, they still maintain stronger positions than the sixth tier, with adjacent tiers showing similar niche strengths. Jining, Linyi, Weifang, and Yantai performed well in terms of resource base and industrial development, but were slightly inferior to the first tier in terms of location conditions, basic guarantees, and construction management, so they were classified as the second tier. Zibo, Tai'an, and Weihai performed moderately in terms of resource base and industrial development but had certain shortcomings in terms of location conditions, basic guarantees, and construction management, so they were classified as the third tier. Rizhao and Zaozhuang in the fourth tier performed relatively poorly in terms of the resource base and industrial development but had certain potential in terms of location conditions, basic guarantees, and construction management, so they were classified as the fourth tier. Heze and Liaocheng in the fifth tier performed poorly in terms of resource base and industrial development, and also had obvious deficiencies in location conditions, basic guarantees, and construction management, but their competitiveness was slightly higher than that of the sixth tier, so they were classified as the fifth tier.

## 4.3 Niche overlap of rural sports tourism competitiveness

The weighted decision matrix $X'$ is standardized, and the niche overlap matrix is calculated using Formula (7) (Table 4). According to niche overlap theory, entities that share resources face competitive pressure from others while simultaneously exerting pressure on their competitors [60]. In the analysis of niche overlap, the row (column) and ±1 adjustment methods are used to calculate the net pressure of rural sports tourism in each city, reflecting its relative position in the competitive landscape. By adjusting the row (column) and ±1, it is possible to more accurately measure the competitive pressure released by each city and the pressure it bears, thereby revealing the dynamic balance of the competitive relationship. In this study, the total pressure released by rural sports tourism in each city is calculated by summing the values in each row of the niche overlap matrix, with adjustments of ±1. Similarly, the competitive pressure exerted by external factors on rural sports tourism in each city is determined by summing the values in each matrix column, also adjusted by ±1. The net pressure experienced by rural sports tourism in each town (as shown in Fig 4) is then derived from the difference between the pressure exerted on the external environment and the pressure received internally. When calculating the total pressure released to the outside world and the competitive pressure from the outside world, subtracting 1 is to exclude the overlap between itself and itself (usually 1) to avoid repeated counting.

According to Table 4, the calculated niche overlap is 1.3438, indicating that the overlap of rural sports tourism in the 16 cities of Shandong Province is relatively serious, indicating that the similarity of the competitive factors of rural sports tourism in each city is relatively high, and there is fierce competition [61]. At the same time, combined with Fig 4, it can be seen that the net pressure of Tai'an, Weihai, Zibo, Jining, Linyi, Weifang, Yantai, Jinan, and Qingdao is positive, indicating that these nine cities are under great external competitive pressure and have less competitiveness, and are in a dominant position in the competition pattern. Among them, Qingdao, Jinan, and Yantai have mutual pressure from the competition of factors, so the competition is the most intense; the net pressure of Zaozhuang, Rizhao, Liaocheng, Binzhou, Dongying, Dezhou, and Heze is negative, and the pressure they are under is much greater than the external competitive pressure, and Dongying and Dezhou are ranked first and second in total pressure, indicating that Dongying and Dezhou are in the weakest and second weakest positions;

**Table 4. Niche overlap matrix of rural sports tourism competitiveness in cities of Shandong Province.**

| City | Zaozhuang | Rizhao | Liaocheng | Binzhou | Tai'an | Dongying | Weihai | Dezhou | Heze | Zibo | Jining | Linyi | Weifang | Yantai | Jinan | Qingdao |
|---|---|---|---|---|---|---|---|---|---|---|---|---|---|---|---|---|
| Zaozhuang | 1 | 0.9792 | 1.5862 | 1.9344 | 0.8266 | 3.287 | 0.7225 | 1.8401 | 1.192 | 0.801 | 0.6072 | 0.6246 | 0.649 | 0.5253 | 0.4356 | 0.4429 |
| Rizhao | 1.0213 | 1 | 1.6199 | 1.9755 | 0.8442 | 3.3569 | 0.7379 | 1.8792 | 1.2173 | 0.818 | 0.6201 | 0.6379 | 0.6628 | 0.5364 | 0.4448 | 0.4524 |
| Liaocheng | 0.6304 | 0.6173 | 1 | 1.2195 | 0.5211 | 2.0723 | 0.4555 | 1.16 | 0.7515 | 0.5049 | 0.3828 | 0.3938 | 0.4092 | 0.3311 | 0.2746 | 0.2792 |
| Binzhou | 0.517 | 0.5062 | 0.82 | 1 | 0.4273 | 1.6993 | 0.3735 | 0.9512 | 0.6162 | 0.4141 | 0.3139 | 0.3229 | 0.3355 | 0.2715 | 0.2252 | 0.229 |
| Tai'an | 1.2098 | 1.1846 | 1.919 | 2.3402 | 1 | 3.9766 | 0.8741 | 2.2261 | 1.4421 | 0.969 | 0.7346 | 0.7557 | 0.7852 | 0.6355 | 0.527 | 0.5359 |
| Dongying | 0.3042 | 0.2979 | 0.4826 | 0.5885 | 0.2515 | 1 | 0.2198 | 0.5598 | 0.3626 | 0.2437 | 0.1847 | 0.19 | 0.1975 | 0.1598 | 0.1325 | 0.1348 |
| Weihai | 1.3841 | 1.3553 | 2.1954 | 2.6773 | 1.144 | 4.5494 | 1 | 2.5468 | 1.6498 | 1.1086 | 0.8404 | 0.8645 | 0.8983 | 0.727 | 0.6029 | 0.6131 |
| Dezhou | 0.5435 | 0.5321 | 0.862 | 1.0513 | 0.4492 | 1.7864 | 0.3927 | 1 | 0.6478 | 0.4353 | 0.33 | 0.3395 | 0.3527 | 0.2855 | 0.2367 | 0.2407 |
| Heze | 0.8389 | 0.8215 | 1.3307 | 1.6228 | 0.6935 | 2.7576 | 0.6061 | 1.5437 | 1 | 0.6719 | 0.5094 | 0.524 | 0.5445 | 0.4407 | 0.3654 | 0.3716 |
| Zibo | 1.2485 | 1.2225 | 1.9804 | 2.4151 | 1.032 | 4.1039 | 0.9021 | 2.2974 | 1.4882 | 1 | 0.7581 | 0.7798 | 0.8103 | 0.6558 | 0.5438 | 0.553 |
| Jining | 1.6469 | 1.6127 | 2.6124 | 3.1858 | 1.3613 | 5.4135 | 1.1899 | 3.0305 | 1.9631 | 1.3191 | 1 | 1.0287 | 1.0689 | 0.8651 | 0.7174 | 0.7295 |
| Linyi | 1.601 | 1.5677 | 2.5395 | 3.0969 | 1.3234 | 5.2625 | 1.1567 | 2.9459 | 1.9084 | 1.2823 | 0.9721 | 1 | 1.0391 | 0.8409 | 0.6974 | 0.7091 |
| Weifang | 1.5408 | 1.5087 | 2.444 | 2.9805 | 1.2736 | 5.0646 | 1.1132 | 2.8351 | 1.8366 | 1.2341 | 0.9355 | 0.9624 | 1 | 0.8093 | 0.6711 | 0.6825 |
| Yantai | 1.9038 | 1.8642 | 3.0198 | 3.6827 | 1.5737 | 6.2578 | 1.3755 | 3.5031 | 2.2693 | 1.5248 | 1.156 | 1.1891 | 1.2356 | 1 | 0.8293 | 0.8433 |
| Jinan | 2.2958 | 2.248 | 3.6416 | 4.3672 | 1.8977 | 7.5463 | 1.6587 | 0.2367 | 2.7366 | 1.8388 | 1.394 | 1.434 | 1.49 | 1.2059 | 1 | 1.0169 |
| Qingdao | 2.2577 | 2.2107 | 3.5811 | 4.3672 | 1.8662 | 7.421 | 1.6312 | 4.1543 | 2.6911 | 1.8083 | 1.3708 | 1.4102 | 1.4653 | 1.1859 | 0.9834 | 1 |

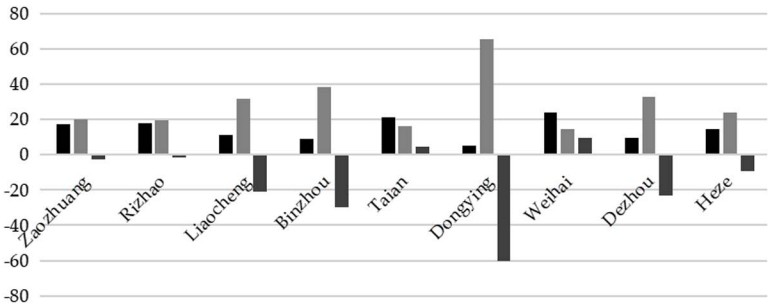

**Fig 4. Competitive pressure on rural sports tourism in municipalities of Shandong province.**

Zaozhuang, Rizhao, Liaocheng, Binzhou, and Heze are in a relatively weak position. It can be seen that although Shandong Province has complex topography, similar resources, spatial and temporal locations, resource markets and other similar factors will lead to fierce competition. Cities with resource advantages, better resource markets, high-quality management and service levels will release relatively greater pressure on other cities, and will be less exposed to external pressure. Other cities will find it difficult to threaten them, and they will be able to occupy a favorable position in the competition. Conversely, cities that do not have an advantage will bear external pressure because they are close to cities with higher competitiveness.

## 5. Research conclusions and theoretical contributions

### 5.1 Key findings

Based on the niche theory, this study constructed a five-level comprehensive evaluation index system, and systematically evaluated the competitiveness of rural sports tourism in 16 cities in Shandong Province, effectively revealing the niche intensity and overlap of rural sports tourism. The main findings of the study are as follows:

**5.1.1 Differences in competitiveness.** The competitiveness of rural sports tourism in various cities in Shandong Province presents six obvious levels, and the resource overlap between cities is relatively serious. The cities with strong competitiveness occupy an advantageous position, while the cities with weak competitiveness are at a disadvantage.

**5.1.2 Hierarchical structure.** The competitiveness of rural sports tourism in various cities in Shandong Province presents six obvious levels, and the resource overlap between cities is relatively serious. The cities with strong competitiveness occupy an advantageous position, while the cities with weak competitiveness are at a disadvantage.

**5.1.3 Reality issues.** All cities in Shandong Province generally face practical problems such as the need to strengthen resource development and utilization, enrich economic development formats, improve location conditions, enhance basic guarantees, promote marketing management and strengthen city-level cooperation.

### 5.2 Theoretical contributions and enlightenment

#### 5.2.1 Comparison with existing research.

1. Deepen the differences in competitiveness dimensions. This study verified the conclusion of Wang & Zhu (2019) [49] on the core role of resource endowment, but further found through the measurement of niche width that the absolute advantage of resource endowment is not completely transformed into competitiveness. When the resource overlap between cities exceeds the threshold [], the difference in competitiveness is narrowed. This finding links the traditional static resource view with the dynamic competition perspective, verifies the view of Glasser JW & Price HJ. (1982) [35] that the development of rural sports tourism requires a "resource combination niche", reveals the boundary conditions of the role of resource endowment, and fills the theoretical gap of the traditional static resource view.

2. The theory expands the hierarchical structure. Compared with the conclusion of four competitiveness levels by Jia et al. (2022) [29], this study identified six competitiveness levels, which not only verified the continuous characteristics of niche differentiation revealed by the progressive separation theory of May RM (1974) [46], but also clarified that resource overlap mainly comes from the differentiated allocation of homogeneous resources, and the overlap effect can be significantly reduced through management coordination. This contrasts with Jaksic-Stojanovic A et al. (2019) [62]'s hypothesis of event homogeneity, revealing that niche segregation can be achieved through management coordination rather than relying solely on resource endowment itself.

3. Quantitative breakthroughs solve practical problems. This study uses the niche theory framework to identify existing resource utilization [51], economic formats [63], and location conditions [64], infrastructure [28], management marketing [3] and municipal cooperation [19] and other common real-life problems, and provided a new explanatory mechanism through quantification, verifying that improving resource utilization is consistent with the ecological niche theory, inefficient resource utilization leads to "vacant" ecological niches; discovered the impact of industrial integration on the expansion of niche width; proposed that management intervention can partially compensate for the poor location; improved the quantitative standard of niche carrying capacity; formed a matching basis for niche and capacity and the evaluation effect of municipal cooperation.

### 5.2.2 Development of the niche theory.

1. Innovation in theoretical application. This study applied the niche theory system to the evaluation of rural sports tourism competitiveness for the first time. The multi-dimensional and comprehensive evaluation system of "resources-industry-location-facilities-management" constructed broke through the limitations of traditional single dimensions or perspectives. Through the design of quantitative indicators and the determination of weights, the synergistic mechanism of multiple factors was revealed, providing a new paradigm for the application of niche theory in the social and economic fields.

2. Expansion of theoretical boundaries. By constructing the "niche overlap-niche intensity" coupling relationship model, it not only verifies the universality of the "resource combination niche" in the context of rural sports tourism, but also clarifies the applicable boundary conditions of the niche theory in this field, filling the gap in the spatial dimension analysis in event tourism research.

### 5.2.3 Implications for tourism competitiveness theory.

1. Supplementation of theoretical framework. This study shows that the niche perspective can effectively explain the role of non-economic factors ignored by traditional competitiveness theory, especially through the "resource synergy-spatial dynamics" two-dimensional analysis framework, which provides a new explanatory path for rural contexts.

2. Expansion of competitiveness research. This study reveals that the multi-dimensional evaluation perspective of the niche theory can effectively supplement the limitation of the existing tourism competitiveness theory that pays insufficient attention to non-economic factors. This theoretical integration provides a new analytical perspective for understanding the competitiveness of rural sports tourism.

3. Methodological contribution. The evaluation system established in this study provides a replicable method tool for tourism competitiveness research, especially the quantitative analysis of resource homogeneity issues, which makes up for the lack of quantitative research on dynamic competition in this field.

## 5.3 Practical significance

### 5.3.1 Consolidate the resource endowment foundation and promote the synergy of rural sports tourism.
According to the niche theory, resource endowment is the core element of rural sports tourism competitiveness,

and the diversity and uniqueness of resources determine the width and overlap of the niche [65]. In order to promote the synergy of rural sports tourism, on the one hand, cities with strong resource endowments (Qingdao, Yantai, etc.) should focus on the sustainable development of resources, avoid over-reliance on a single resource, and enhance the long-term competitiveness of resources through innovation and diversified development. Cities with weak resource endowments (Dongying, Dezhou, etc.) should establish a coordinated development mechanism with surrounding cities with strong resource endowments, form regional cooperation, share resources, and enhance overall competitiveness. On the other hand, policymakers should formulate specific resource protection and development policies to ensure the sustainable use of resources. At the same time, the government can support resource integration and coordinated development in cities with weaker resource endowments by setting up special funds. Practitioners should actively participate in resource development projects, use policy support, and innovate resource utilization methods. Scholars should conduct in-depth research on resource endowment assessment methods to provide scientific basis for policy formulation and practitioners. Through the above measures, the rational development of urban resource endowments and regional coordination will be promoted, the width of ecological niches will be expanded, resource utilization efficiency will be improved, and overall competitiveness will be enhanced.

### 5.3.2 Give full play to the economic effect of the industry and expand the rural sports tourism industry.

Industrial development is an significant support for the competitiveness of rural sports tourism. Industrial agglomeration and business innovation can effectively enhance the competitive advantage of the ecological niche [49]. In order to expand the rural sports tourism industry, on the one hand, cities with better industrial development (such as Jinan and Qingdao) should continue to promote industrial agglomeration and business innovation, and enhance brand influence by holding large-scale events and competitions. On the other hand, cities with weaker industrial development (such as Liaocheng and Binzhou) should increase the scale of the industry through government guidance and social capital investment, learn from the experience of successful regions, and develop projects with local characteristics. On the other hand, policymakers should introduce industrial support policies to encourage industrial agglomeration and business innovation. Practitioners should actively participate in the construction of industrial agglomeration areas and use policy support to develop sports tourism projects with local characteristics. Scholars should conduct in-depth research on the models of industrial agglomeration and business innovation to provide theoretical support for policy makers and practitioners. Through the above measures, the overall competitiveness will be enhanced by promoting industrial agglomeration and business innovation in the city, consolidating the industrial niche and expanding the industrial scale.

### 5.3.3 Improve market location conditions and develop rural sports tourism sources.

Location conditions are significant factors affecting the competitiveness of rural sports tourism. Good location conditions can effectively reduce niche overlap and enhance market competitiveness [63]. In order to develop rural sports tourism sources, on the one hand, regions with better location conditions (Qingdao, Jinan, etc.) should make full use of their advantages in transportation and source markets to further develop potential sources of tourists and attract more tourists by optimizing transportation networks and tourist routes. On the other hand, regions with weaker location conditions (Dezhou, Dongying, etc.) should improve their location advantages by improving transportation infrastructure, cooperate with surrounding areas, and form regional linkages. On the other hand, policymakers should formulate policies for transportation infrastructure construction and tourist route optimization to enhance location advantages. Practitioners should actively participate in transportation infrastructure construction and tourist route development, and use policy support to enhance the attractiveness of the source market. Scholars should conduct in-depth research on the impact of location conditions on the competitiveness of rural sports tourism, and provide scientific basis for policy makers and practitioners. Through the above measures, the overall competitiveness will be significantly improved by improving the location conditions of the urban market, expanding the urban source market and improving the location disadvantages.

**5.3.4 Strengthen infrastructure construction and optimize rural sports tourism environment.** Infrastructure is an significant guarantee for the competitiveness of rural sports tourism. Perfect infrastructure can effectively improve the stability and sustainability of the ecological niche [66]. To optimize rural sports tourism, on the one hand, cities with better infrastructure should continue to consolidate and improve infrastructure construction, optimize the tourism environment, and enhance the tourist experience and overall environmental quality. Cities with weaker infrastructure should improve infrastructure through government financial support and social capital investment, cooperate with surrounding areas, and share resources. On the other hand, policymakers should formulate infrastructure construction policies to improve the level of infrastructure support. Practitioners should actively participate in infrastructure construction, use policy support, and improve the quality of the tourism environment. Sc competitiveness of rural sports tourism and provide theoretical support for policy makers and practitioners. Through the above measures, we will strengthen the construction of urban infrastructure, optimize the sports tourism environment, promote the improvement of basic guarantee levels and make up for the shortcomings of infrastructure, and the overall competitiveness will be significantly enhanced.

**5.3.5 Improve construction management and rural sports tourism services.** The level of construction and management is an significant determinant of the competitiveness of rural sports tourism. Efficient management and services can effectively enhance the functional value of the ecological niche [67]. In order to provide rural sports tourism services, on the one hand, cities with better construction and management (Qingdao, Jinan, etc.) should continue to improve marketing and market supervision, and enhance brand influence by holding large-scale events and competitions. Cities with weaker construction and management (Dezhou, Binzhou, etc.) should improve management by strengthening talent training and digital construction, learn from the experience of successful regions, and improve overall service quality. Policymakers should formulate policies to improve construction management and service levels, improve management efficiency and service quality. Practitioners should actively participate in management and service level improvement projects, and use policy support to improve service quality. Scholars should conduct in-depth research on the model of construction management and service level improvement to provide theoretical support for policy makers and practitioners. By improving the level of urban construction management through the above measures, promoting the improvement of management and service levels and improving management shortcomings, the overall competitiveness will be significantly improved.

## 5.4 Universality of theory and practice

The above suggestions not only provide practical guidance for the development of rural sports tourism in Shandong Province, but also lay the foundation for the theoretical significance of this study and its universality at home and abroad. The index system designed based on the theoretical framework of this study fully considers the resource conditions and development levels of different regions, which can not only provide a scientific basis for the development of rural sports tourism in Shandong Province, but also provide a reference for related research in other regions at home and abroad. For other regions in China, this index system can effectively identify competitive advantages. For example, the Yangtze River Delta region has outstanding performance in industrial agglomeration and brand building due to its developed economy and superior location conditions [49]. This index system can capture its advantages in industrial agglomeration and marketing; the Pearl River Delta region has attracted a large number of tourists through the combination of sports events and cultural activities [68]. This index system can reflect its characteristics of resource integration and business innovation; the central and western regions have huge potential for the development of rural sports tourism based on rich natural resources and unique cultural background [69]. This index system can identify the competitive advantages of resource endowment and cultural characteristics.

This theoretical framework is also applicable to the comparative analysis of international rural sports tourism competitiveness. By comparing the development models of rural sports tourism in different countries, we can find that the United States

has significant advantages in resource base and industrial development, and has promoted the development of rural tourism through large-scale sports events (such as marathons and cycling races) [47]; France has attracted a large number of international tourists by integrating cultural heritage protection and sports activities [63]; Italy has developed rural sports tourism featuring hiking and mountain biking, relying on its rich natural resources and historical culture [70]. These international experiences show that the niche theory and the evaluation index system designed are universal in analyzing the competitiveness of rural sports tourism, and can provide theoretical support for the development of rural sports tourism in different countries and regions.

## 6. Limitations and future research

### 6.1 Study limitations

Although the indicator system of this study has strong universality, it still has certain limitations. First, the data source is mainly limited to Shandong Province, and other provinces and regions are not included as empirical comparison cases. Second, this study mainly uses quantitative analysis, and there are potential subjective biases in some indicators that require weighted processing.

### 6.2 Future research directions

With the development of rural sports tourism, the exploration of new competitiveness evaluation indicators and methods will become an important direction for future research. It is recommended that when applying this study, the data scope should be further expanded to cover more regions and countries, so as to achieve the difference in the competitiveness of rural sports tourism in different regions and the application of niche theory in other types of tourism research. At the same time, the local resource endowment, economic development level, cultural characteristics, and qualitative data such as resource quality scores and tourist satisfaction surveys should be considered to appropriately adjust the evaluation system and enhance the comprehensiveness of the research through a combination of quantitative and qualitative analysis methods. At the same time, it is recommended that countries refer to the policy recommendations proposed in this article when formulating rural sports tourism policies, and optimize them in combination with their own actual conditions. These research approaches can not only enhance the in-depth understanding of rural sports tourism, but also help to formulate more effective and targeted policy frameworks, and provide a new paradigm for the application of niche theory in the field of rural sustainable development.

## Supporting information

**S1 Fig. Evaluation index system of urban-rural sports tourism competitiveness.**
(XLSX)

## Acknowledgments

Data provided by various statistical yearbooks (2023 edition) published in 16 city regions of Shandong Province, the Department of Culture and Tourism of Shandong Province, the Department of Natural Resources of Shandong Province, the Shandong Provincial Bureau of Statistics, the municipal Culture and Tourism Bureaus, and other relevant departments contributed to the further analysis of this study.We would like to thank Dr. Jianda Kong for helping to revise and improve this manuscript.

This work was supported by the School of Sports Science, Qufu Normal University.

## Author contributions

**Conceptualization:** Yin Lijun, Gao Xueting, TAI Pengfei, Cao Li.

**Data curation:** Yin Lijun, Tai Pengfei.

**Formal analysis:** Yin Lijun, Gu Qian, Gao Xueting.

**Investigation:** Gu Qian, Tai Pengfei.

**Methodology:** Yin Lijun.

**Supervision:** Cao Li.

**Validation:** Yin Lijun, Gao Xueting, Tai Pengfei.

**Visualization:** Gu Qian, Tai Pengfei.

**Writing – original draft:** Yin Lijun, Gu Qian, Gao Xueting, Tai Pengfei.

**Writing – review & editing:** Yin Lijun, Gu Qian, Gao Xueting, Tai Pengfei, Cao Li.

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
