## [Decision Letter · Decision Letter 0]

14 Dec 2024

PONE-D-24-42459Rural Sports Tourism Competitiveness in Shandong ProvincePLOS ONE

Dear Dr. Li,

Thank you for submitting your manuscript to PLOS ONE. After careful consideration, we feel that it has merit but does not fully meet PLOS ONE’s publication criteria as it currently stands. Therefore, we invite you to submit a revised version of the manuscript that addresses the points raised during the review process.

We look forward to receiving your revised manuscript.

Kind regards,

Yihao Li, Doctor

Academic Editor

PLOS ONE

Journal Requirements:

2. We note that your Data Availability Statement is currently as follows: All relevant data are within the manuscript and its Supporting Information files

Reviewers' comments:

Reviewer's Responses to Questions

**Comments to the Author**

1. Is the manuscript technically sound, and do the data support the conclusions?

Reviewer #1: Partly

Reviewer #2: Partly

Reviewer #3: Partly

Reviewer #4: Partly

2. Has the statistical analysis been performed appropriately and rigorously? 

Reviewer #1: No

Reviewer #2: Yes

Reviewer #3: Yes

Reviewer #4: N/A

3. Have the authors made all data underlying the findings in their manuscript fully available?

Reviewer #1: No

Reviewer #2: Yes

Reviewer #3: No

Reviewer #4: No

4. Is the manuscript presented in an intelligible fashion and written in standard English?

Reviewer #1: Yes

Reviewer #2: Yes

Reviewer #3: Yes

Reviewer #4: Yes

5. Review Comments to the Author

Reviewer #1: Comment 1:The author did not fully demonstrate the reliability of the data. First of all, the data cleaning process was not provided. Are there missing values and outliers? Secondly, the authors use the least square method but do not inform whether the normality test for the result residuals has been carried out, which is essential.

Comment 2:With strong limitations, this paper only discusses the current situation of rural sports tourism in Shandong Province, China, and the conclusion is based on 16 cities in Shandong Province, without discussing whether it can be promoted in other regions or even countries. The practical significance of this paper is not strong, so it is suggested that the discussion content can be modified from the perspective of the applicability of the method.

Comment 3:This is not a social science article in the full sense, the author uses statistical methods of data analysis, but there is no strict logic in its use. Please supplement the analysis of statistical significance and the robust test of the results. Are the results reliable enough to be interpreted?

Comment 4:Why is there no reference in the first paragraph of introduction? For example: (Lines 8-11) How is the conclusion reached? (Line 24) How can the author confirm that there are few studies on the competitiveness of sports tourism? The whole introduction is more like an undergraduate literature review, without any logic, and does not explain clearly what the author wants to express. At the same time, it is suggested to introduce more references.

This paper has a certain innovation, the establishment of a relatively comprehensive competitiveness evaluation index system, and the use of data analysis method is relatively complete. However, there are many deficiencies in data processing, and the generalization of the conclusions is insufficient. The interpretation of the analysis results only stops at the description of the phenomenon and lacks in-depth discussion on the policy level. Recommended to accept after Major Revision.

Reviewer #2: The manuscript addresses an important and relevant topic—evaluating rural sports tourism competitiveness in Shandong Province using ecological niche theory and quantitative methods. While the research design and methodology are technically sound, certain areas require clarification and improvement to ensure the study's impact and reproducibility.

Here are my suggestions:

1. Please ensure that all data underlying the findings are made fully available and stored in an open-access repository. This will enhance the transparency and reproducibility of the study.

2. Expand the discussion of the theoretical significance of your findings and their applicability beyond Shandong Province. A comparative analysis with international examples or other regions would strengthen the manuscript’s global perspective.

3. Provide clearer justification for the thresholds and classifications used in the competitiveness hierarchy and niche overlap analysis. This will improve the credibility of your results.

4. The manuscript requires further language editing to improve the clarity and fluency of the writing, making it more accessible to an international audience. Technical terms should be better explained for non-expert readers.

5. The literature review should be separated from the introduction and presented as a distinct section. Currently, it is embedded within the introduction, which makes the flow of the paper less structured. Ensure the literature review is comprehensive and focused on relevant studies.

6. The conclusion section does not fully align with standard journal formatting. Please restructure it to summarize key findings, limitations, and potential directions for future research, following the common conventions for academic papers.

Reviewer #3: Summary and Overall Impression

This manuscript attempts to analyze the competitive relationships among rural sports tourism cities in Shandong Province by applying niche theory. Employing a least squares optimization model to determine both subjective and objective weights for the indicators, the authors develop a five-tier evaluation index system. The study examines sixteen cities in Shandong Province, measuring their niche intensity and overlap to assess competitive dynamics.

While the research framework demonstrates a structured methodological approach, there are several aspects that could benefit from further clarification. First, although the study focuses on "rural sports tourism" the distinctive characteristics of rural sports could be more thoroughly addressed. For instance, the five dimensions adopted by the authors - resource base, industrial development, location conditions, basic guarantees, and construction management - are comprehensive but might be enriched by incorporating specific indicators that reflect the unique features of rural sports tourism.

Second, some methodological details would benefit from additional explanation to enhance reproducibility, particularly regarding the process of weight determination and niche overlap calculations. The ±1 adjustments in the calculation process would be more convincing with further explanations. Additionally, while the subjective index contributes significantly to the final composite index, the process of obtaining these subjective indexes could be more thoroughly documented, including the evaluation criteria and procedures. Such additional information would strengthen the study's methodological foundation.

A more critical issue lies in the disconnect between the analytical framework and the proposed recommendations. The authors suggest several approaches for improving destination competitiveness; however, these recommendations do not appear to be directly derived from either the tier evaluation index analysis or the weights analysis. This raises questions about the logical flow from findings to conclusions. It could be much clearer if the authors add more information with respect to the derivation of the suggestions.

Strength:

1. The application of Niche Theory provides an innovative theoretical framework for analyzing tourism competitiveness among different cities, offering a fresh ecological perspective to tourism research.

2. The study presents a comprehensive quantitative analysis with detailed descriptions of objective indexes and systematic evaluation procedures across multiple dimensions.

3. The research makes a significant practical contribution by thoroughly examining the competitive landscape of rural sports tourism across 16 cities in Shandong Province, providing valuable regional insights.

Weakness:

1. The methodology lacks sufficient transparency in several key processes, particularly in the calculation of subjective indexes and niche overlap adjustments, which significantly limits the study's reproducibility.

2. The evaluation framework employs generic tourism indicators without adequately addressing the unique characteristics and requirements specific to rural sports tourism development.

3. There exists a notable disconnect between the analytical findings and the proposed recommendations, as the suggestions for improvement do not appear to be directly derived from the study's quantitative results.

Major Issues

1. A significant methodological concern arises from the omission of AHP calculation details �Line 90-95�. While the authors acknowledge AHP as their subjective weighting method, the detailed documentation of AHP implementation, including expert selection criteria, pairwise comparison matrices, and consistency checks, is fundamental for ensuring research reliability and reproducibility.

2. The presentation of the five-dimensional evaluation framework would benefit from a clearer theoretical development process. Currently, the authors directly present these dimensions without sufficient theoretical scaffolding (Figure 1) The readers cannot follow the logical progression from previous literature to the current framework. A clear demonstration of how these dimensions evolved from existing research to specifically address rural sports tourism would strengthen the theoretical foundation.

3. A methodological concern lies in the measurement of tourism resource quality �Line 161-162�. The authors' approach of "counting the types of available resources" appears oversimplified. This quantitative method seems to assume that quantity directly correlates with quality. The authors should either provide empirical evidence from previous studies to support this measurement approach or consider incorporating more comprehensive quality assessment methods.

4. As for the calculation of “The popularity of rural sports tourism resources (X13)”(Line 164-166), why did the author set the weights to 10 (for national rural sports tourism resources) and 8 (for provincial rural sports tourism resources)? A similar concern also applies to the calculation of X23 (Line 174-178).

5. The authors' methodology of adjusting the row �columns as well� sums by ±1 �Line 346-355�requires further explanation and justification. As presented, readers may not understand when to add or subtract 1, why this adjustment is necessary, or what it represents in the context of pressure measurement.

6. Another major concern lies in the disconnect between the study's analytical findings and its recommendations. While the authors have conducted sophisticated analyses of city competitiveness rankings and niche relationships, their recommendations (such as developing natural resources, expanding industry, and strengthening infrastructure) appear to be generic suggestions without clear linkage to their empirical findings. The authors could establish a stronger logical connection by demonstrating how each recommendation specifically relates to their analytical results.

Minor Issues

1. Formula(2) need more clarification since it is somewhat different from the standard MacArthur model. (e.g., what do 'p', 'o', and 'R' represent)

2. The authors should report the specific β value used in their least squares method for combining subjective and objective weights.

3. The visualization of results needs improvement. For example, in Figure 2, the bar chart uses colors that are too similar to distinguish different categories.

Reviewer #4: General comment – First, I appreciate the opportunity to review this manuscript, which explores the competitiveness of rural sports tourism in Shandong Province through the lens of niche theory. This is a timely and relevant topic, given the crucial role of rural sports tourism in promoting the rural sports industry and achieving rural revitalisation.

While the manuscript presents an interesting perspective by developing a five-tier evaluation index system and applying the least squares optimisation model, I believe there are areas that require substantial improvement to enhance the clarity, coherence, and impact of the study. The structure of the manuscript could be refined to better articulate the research objectives and findings. Additionally, the theoretical foundation needs further development to establish a stronger rationale for the use of niche theory and its application to the evaluation of rural sports tourism competitiveness.

The methodology section would also benefit from greater detail and justification, particularly regarding the determination of subjective and objective weights for the indicators and the robustness of the analytical approach. Strengthening these aspects would significantly contribute to the credibility and practical relevance of the study.

I hope the authors will find these comments constructive and useful in improving the manuscript for future consideration.

Comment 1 – The current title, "Rural Sports Tourism Competitiveness in Shandong Province," suggests a case study focused exclusively on Shandong Province, which may inadvertently limit the perceived generalisability of the study's findings. To better reflect the broader implications and applicability of the research, the title could be revised to emphasise the theoretical framework, methodology, or insights derived from the study. For instance, incorporating terms like "Framework," "Model," or "Evaluation" could highlight the study's contribution beyond a specific geographic context.

Comment 2 – The Introduction section requires significant improvements to enhance its clarity and comprehensiveness.

The second paragraph of the Introduction summarises previous research on tourism destination and sports tourism competitiveness. However, the discussion is overly concise and lacks sufficient detail to establish a solid foundation for the study. For example, the cited studies are briefly mentioned without adequately elaborating on their methodologies, key findings, or limitations. Expanding this section to provide a deeper discussion of the existing literature would better contextualise the research. Furthermore, incorporating more recent and region-specific studies on rural sports tourism competitiveness would demonstrate the relevance of the research and situate it within the broader academic discourse. This elaboration would strengthen the theoretical background and provide a clearer connection between the reviewed literature and the current study.

The research gap is unclear and requires explicit articulation. While the manuscript highlights that studies on rural sports tourism competitiveness are relatively limited, it does not sufficiently explain how the current study addresses these gaps or contributes to advancing the field. To strengthen this aspect, the authors should clearly identify the shortcomings in previous research, such as the lack of comprehensive evaluation frameworks or the limited focus on rural sports tourism, and explicitly state how the proposed five-tier evaluation index system and the use of niche theory provide a novel and meaningful contribution. This would clarify the rationale for conducting the study and emphasise its significance.

By addressing these issues, the Introduction would provide a more robust justification for the research and improve the reader's understanding of its objectives and potential contributions.

Comment 3 – The Introduction would benefit from a clearer justification for focusing specifically on Shandong Province. While rural revitalisation is highlighted as a national priority, the manuscript does not adequately explain why this province was chosen as the study context. Shandong’s rich cultural heritage, diverse natural landscapes, and historical significance as a hub for tourism and economic development make it a compelling case for examining rural sports tourism competitiveness. Explicitly linking these characteristics to the research objectives would enhance the contextual relevance. Furthermore, discussing how findings from Shandong might provide insights or serve as a model for other regions would strengthen the study’s broader applicability. Without this justification, the choice of Shandong risks appearing arbitrary, potentially weakening the manuscript’s overall contribution.

Comment 4 – The Theoretical Basis section would benefit from a clearer distinction between the theoretical foundations and the methodological approaches to improve clarity and organisation. Currently, the discussion of niche theory as an ecological framework is well-developed, providing a solid foundation for understanding the theoretical underpinnings of the study. However, the subsequent introduction of formulas and their application to rural sports tourism begins to blend theoretical explanation with methodological details, potentially confusing readers about the purpose of this section.

To address this, the theoretical basis should focus exclusively on explaining the key concepts of niche theory—such as niche strength and niche overlap—in the context of rural sports tourism. This would include a conceptual explanation of how these constructs relate to the competitiveness of cities within the rural sports tourism ecosystem. Discussions about their ecological significance, relevance to tourism studies, and alignment with the research objectives should remain in this section.

The formulas and calculations, along with their application to the evaluation of rural sports tourism, should be moved to the Materials and Methods section. This separation would not only improve the structure of the manuscript but also allow for a more detailed and focused explanation of the methodological approach in its appropriate section. By making this distinction, the manuscript would present a clearer narrative, ensuring that readers can more easily follow the progression from theory to practical application.

Comment 5 – The Methods section would benefit from a more detailed explanation of the AHP calculation steps, as the lack of citations or elaboration on the methodology leaves readers unable to fully understand or replicate the process. While the manuscript acknowledges that the Analytic Hierarchy Process (AHP) is context-dependent, this does not justify omitting either a brief explanation of the calculation steps or a citation to guide readers toward relevant resources.

To address this, the authors should either provide a concise overview of the key steps involved in AHP—such as constructing a pairwise comparison matrix, normalising the matrix, and calculating consistency—or cite authoritative sources that detail the procedure. This would ensure transparency and enable readers to assess the robustness of the subjective weighting method used. Additionally, clarifying how context-specific factors influenced the application of AHP in this study (e.g., the criteria used for pairwise comparisons and the selection of experts) would enhance the methodological clarity and reliability of the results. Without this information, readers may question the validity of the subjective weighting process and its integration with objective methods, potentially undermining the credibility of the evaluation framework.

Comment 6 – The Methods section provides a detailed description of the methodologies used, including the information entropy method, the least squares optimisation model, and the AHP approach. However, as a reviewer, I find it challenging to fully understand the methodological framework due to insufficient contextual explanation and the absence of citations to foundational literature. While the mathematical formulas are presented, the rationale for selecting these methods and their suitability for achieving the research objectives are not clearly articulated. This lack of clarity could make it difficult for readers to assess the robustness and relevance of the methodology.

To improve this section, the authors should provide a concise overview of why these specific methods were chosen and how they address the research questions. Additionally, they should include citations to standard references for these methodologies to guide readers who may not be familiar with them. For example, references explaining the application of the AHP approach or the entropy method in similar contexts would enhance the credibility and transparency of the study. Furthermore, a step-by-step explanation of how the methods were implemented in the specific case of rural sports tourism in Shandong Province would help readers better understand the process and evaluate the results. Without these improvements, the methodology section risks being inaccessible to readers who are not experts in these techniques.

Comment 7 – The Discussion section requires significant revision to adhere to standard academic conventions, enhance its scholarly impact, and provide clear implications.

First, the structure of the discussion does not follow the standard academic tradition. Typically, a discussion section should systematically address the key findings, compare them with existing literature, discuss their implications, and outline limitations and future research directions. However, this section appears more like a series of recommendations or strategies without sufficiently linking them to the study's results. A more structured approach is needed, beginning with an explicit interpretation of the findings in relation to the research objectives and followed by a critical comparison with previous studies.

Second, the discussion does not adequately compare the results of this study with findings from existing literature. For instance, while the study highlights the importance of resource endowment and infrastructure, there is no mention of how these findings align or contrast with prior research on rural tourism competitiveness. Including such comparisons would not only situate the study within the broader academic discourse but also strengthen its contribution by demonstrating how it confirms, extends, or challenges existing knowledge.

Third, the section lacks explicit implications. While some practical suggestions are scattered throughout, the implications for policymakers, practitioners, and academics are not clearly articulated. For instance, what specific policies should be enacted to support rural sports tourism development in Shandong? How can industry stakeholders utilise the study's findings to improve competitiveness? Furthermore, theoretical implications, such as how niche theory contributes to understanding tourism competitiveness, are entirely absent. These implications should be systematically discussed to enhance the relevance and impact of the study.

Comment 8 – The Conclusion section lacks the standard academic structure needed to effectively summarise and contextualise the study. While it briefly discusses variations in competitiveness and proposes strategies for improvement, it fails to clearly link these findings to the study’s objectives or highlight its theoretical and practical contributions. The section does not adequately articulate the implications for policymakers or stakeholders, nor does it discuss how the study advances the application of niche theory in tourism research. Furthermore, it omits critical elements such as the study’s limitations and directions for future research, which are essential for transparency and academic rigour. To improve, the conclusion should concisely summarise the main findings, emphasise their implications, and include a discussion of limitations and potential areas for future investigation.

6. PLOS authors have the option to publish the peer review history of their article (what does this mean? ). If published, this will include your full peer review and any attached files.

**Do you want your identity to be public for this peer review?** For information about this choice, including consent withdrawal, please see our Privacy Policy .

Reviewer #1: No

Reviewer #2: No

Reviewer #3: No

Reviewer #4: No

---

## [Author Response · Author response to Decision Letter 1]

22 Jan 2025

Dear Editors:

Combined with your requirements, we have adjusted the format of the manuscript, uploaded the dataset and associated it with the ORCID iD , and revised the article based on the comments of the reviewers. Thank you very much.

Reviewer #1 :

I am most grateful to you for your valuable suggestions regarding this paper. I have used the highlighting function to indicate the revisions made to the manuscript.

Introduction and Literature review,

Comment 1

1. The author did not fully demonstrate the reliability of the data. First of all, the data cleaning process was not provided. Are there missing values and outliers? Secondly, the authors use the least square method but do not inform whether the normality test for the result residuals has been carried out, which is essential.

Response: We appreciate the reviewer’s suggestion. On the one hand, we have reconfirmed the data cleaning process of rural sports tourism-related data in 16 cities in Shandong Province, including checking whether there are missing values and outliers in the data set, unifying the data format and unit, checking whether there are duplicate records in the data set, and uploading all the confirmed relevant data. On the other hand, we have added the normality test of the residual results in the text to ensure the reliability of the model and the accuracy of the analysis results. (Please see lines 333-338, 436-437)

Comment 2

2. With strong limitations, this paper only discusses the current situation of rural sports tourism in Shandong Province, China, and the conclusion is based on 16 cities in Shandong Province, without discussing whether it can be promoted in other regions or even countries. The practical significance of this paper is not strong, so it is suggested that the discussion content can be modified from the perspective of the applicability of the method.

Response: In response to the review comments, we further revised the paper to enhance its practical significance and universality. First, we clarified the limitations of the study and pointed out that the conclusions are mainly applicable to Shandong Province and may not be directly generalized to other regions or countries. Secondly, from the perspective of method applicability, we added a discussion on the applicability of the method in other domestic regions and in the comparative analysis of international rural sports tourism competitiveness. Finally, we further emphasized the theoretical significance and practical value of the study in the conclusion section and proposed directions for future research. (Please lines 802-834, 848-853)

Comment 3

3. This is not a social science article in the full sense, the author uses statistical methods of data analysis, but there is no strict logic in its use. Please supplement the analysis of statistical significance and the robust test of the results. Are the results reliable enough to be interpreted?

Response: In response to the review comments, we have supplemented the data and process of statistical significance analysis and robustness test, including: using Shapiro-Wilk test to verify the normality of residuals, verifying the significance of the model through goodness of fit and F test; verifying the robustness of the model through weight sensitivity analysis, data perturbation test and subsample analysis. We will upload the relevant calculation code and detailed data. (Please see 322-367,436-449)

Comment 4

4. Why is there no reference in the first paragraph of introduction? For example: (Lines 8-11) How is the conclusion reached? (Line 24) How can the author confirm that there are few studies on the competitiveness of sports tourism? The whole introduction is more like an undergraduate literature review, without any logic, and does not explain clearly what the author wants to express. At the same time, it is suggested to introduce more references.

Response: In response to the review comments, we have comprehensively revised the introduction, specifically adding necessary citations, supplementing literature to support the conclusions, comprehensively combing the literature to enhance logic and clarity, and supplementing the literature on rural sports tourism competitiveness to elaborate on the less studied confirmation process. (Please see lines 28, 34, 65-70, 85-107, 111-113)

I would like to express my gratitude once more for your invaluable contribution to the enhancement of this manuscript. Thank you!

Reviewer #2 :

I am most grateful to you for your valuable suggestions regarding this paper. I have used the highlighting function to indicate the revisions made to the manuscript.

1. Please ensure that all data underlying the findings are made fully available and stored in an open-access repository. This will enhance the transparency and reproducibility of the study.

Response: Thank you for posing this question. We appreciate your suggestion. In response to your suggestion, we have uploaded all the research data to ensure data transparency and reproducibility.

2. Expand the discussion of the theoretical significance of your findings and their applicability beyond Shandong Province. A comparative analysis with international examples or other regions would strengthen the manuscript’s global perspective.

Response: In response to the review comments, we have added relevant explanations in the introduction, including international discussions supported by literature, comparative analysis of other regions in China, and suggestions for future international comparative research to make up for the lack of comparative analysis with international samples. (Please see lines 34-36, 802-834, 848-864)

3. Provide clearer justification for the thresholds and classifications used in the competitiveness hierarchy and niche overlap analysis. This will improve the credibility of your results.

Response: In response to the review comments, we have supplemented the thresholds and detailed reasons for the classification of competitiveness hierarchy and niche overlap analysis in the corresponding section, including the explanation of the competitiveness hierarchy and niche overlap divided by cluster analysis and MacArthur model, and provided specific classification standards and thresholds. (Please see lines 582-583, 587-591, 607-620)

4. The manuscript requires further language editing to improve the clarity and fluency of the writing, making it more accessible to an international audience. Technical terms should be better explained for non-expert readers.

Response: We appreciate your attention to the linguistic aspects of our manuscript. We understand that clear and concise communication is essential for scientific writing. In response. We have made comprehensive revisions to the paper based on your suggestions, focusing on solving problems such as colloquial vocabulary and sentence patterns, tense and sentence components, logical coherence, and grammatical errors. Specific measures include: replacing colloquial vocabulary and sentence patterns, unifying tenses and supplementing sentence components, adding transitional vocabulary to enhance logical coherence, and correcting grammatical errors. At the same time, we have cited basic literature and explained in more detail key terms such as niche theory, niche strength, niche overlap, and least squares method to ensure that non-professional readers can understand. (Please see lines 10-23, 25-34, 37-53, 96-104)

5. The literature review should be separated from the introduction and presented as a distinct section. Currently, it is embedded within the introduction, which makes the flow of the paper less structured. Ensure the literature review is comprehensive and focused on relevant studies.

Response: We appreciate the reviewer’s suggestion. We separated the literature review from the introduction and listed it as Part 2. We also expanded and optimized the content to ensure that it fully covers relevant literature such as rural tourism competitiveness, sports tourism competitiveness, and the application of niche theory in tourism research. At the same time, we highlighted the shortcomings of existing research and the innovation of this study. (Please see lines 65-70, 85-107, 111-113)

6. The conclusion section does not fully align with standard journal formatting. Please restructure it to summarize key findings, limitations, and potential directions for future research, following the common conventions for academic papers.

Response: In response to the review comments, we have reorganized the conclusion section according to the common practice of academic papers, and divided it into four parts: conclusion, suggestions, theoretical significance and extension, and discussion. The conclusion summarizes the main results of the study, and the suggestions propose measures to enhance the competitiveness of rural sports tourism based on the research results. The theoretical significance and extension clarify the theoretical significance of this study and its applicability at home and abroad. The discussion compares the results of this study with existing research results and explains the shortcomings and future research directions. (Please see lines 668-864)

I would like to express my gratitude once more for your invaluable contribution to the enhancement of this manuscript. Thank you!

Reviewer #3:

I am most grateful to you for your valuable suggestions regarding this paper. I have used the highlighting function to indicate the revisions made to the manuscript.

Major Issues,

1. A significant methodological concern arises from the omission of AHP calculation details �Line 90-95�. While the authors acknowledge AHP as their subjective weighting method, the detailed documentation of AHP implementation, including expert selection criteria, pairwise comparison matrices, and consistency checks, is fundamental for ensuring research reliability and reproducibility.

Response: We appreciate your suggestion . Based on your suggestions, we have supplemented the calculation details of AHP (Analytic Hierarchy Process), including the expert selection criteria, the construction of the pairwise comparison matrix, and the consistency check steps, and uploaded the relevant data to ensure the reliability and repeatability of the research. (Please see lines 244-274)

2. The presentation of the five-dimensional evaluation framework would benefit from a clearer theoretical development process. Currently, the authors directly present these dimensions without sufficient theoretical scaffolding (Figure 1) The readers cannot follow the logical progression from previous literature to the current framework. A clear demonstration of how these dimensions evolved from existing research to specifically address rural sports tourism would strengthen the theoretical foundation.

Response: We are very grateful for your suggestions. Based on your suggestions, we supplemented the theoretical development process of the five-dimensional evaluation framework, elaborated in detail how the five dimensions of resource base, industrial development, location conditions, basic guarantees and construction management have gradually evolved from existing research, and adjusted and expanded them specifically for rural sports tourism. We combined the relevant research on rural tourism, sports tourism and tourism destination management to show the theoretical sources and logical progression of the relevant dimensions. (Please see lines 158-168)

3. A methodological concern lies in the measurement of tourism resource quality �Line 161-162�. The authors' approach of "counting the types of available resources" appears oversimplified. This quantitative method seems to assume that quantity directly correlates with quality. The authors should either provide empirical evidence from previous studies to support this measurement approach or consider incorporating more comprehensive quality assessment methods.

Response: Thank you for pointing out the need for clarification. We understand your concern about the simplification of the "calculating available resource types" method. In this article, this method is based on the resource diversity theory, assuming that the richness of resource types can enhance the attractiveness of tourist destinations, and is simple and operational in actual operation, suitable for large-scale empirical research. Although this method has certain rationality in the research context of this article, we fully agree with the experts' suggestions and point out in the discussion section that future research will further optimize this method and introduce more comprehensive quality evaluation indicators such as resource quality scores and tourist satisfaction surveys to improve the scientificity and reliability of the research. (See lines 857-859)

4. As for the calculation of “The popularity of rural sports tourism resources (X13)”(Line 164-166), why did the author set the weights to 10 (for national rural sports tourism resources) and 8 (for provincial rural sports tourism resources)? A similar concern also applies to the calculation of X23 (Line 174-178).

Response: We are very grateful for the problem you pointed out. The current weight setting (10 points for national level and 8 points for provincial level) is based on the influence of resources, policy support and experience reference of similar studies at home and abroad, and has certain rationality and operability. National resources are given higher weights due to their higher visibility, policy support and market recognition. In future research, we will consider using the analytic hierarchy process (AHP) or expert scoring method to further optimize the weight setting, and introduce more evaluation dimensions (such as resource quality and tourist satisfaction) to improve the scientificity and comprehensiveness of the research. (Please see lines 158-164)

5. The authors' methodology of adjusting the row �columns as well� sums by ±1 �Line 346-355�requires further explanation and justification. As presented, readers may not understand when to add or subtract 1, why this adjustment is necessary, or what it represents in the context of pressure measurement.

Response: We are very grateful for your valuable comments. Based on your suggestions, we have explained and demonstrated the row (column) and ±1 adjustment methods in detail, supplementing their background, purpose and specific steps. By calculating the row and column sums of the niche overlap matrix and subtracting 1 to exclude the overlap of itself, we can more accurately measure the net pressure of rural sports tourism in each city and reveal its relative position in the competitive landscape. (Please see lines 625-630, 637-640)

6. Another major concern lies in the disconnect between the study's analytical findings and its recommendations. While the authors have conducted sophisticated analyses of city competitiveness rankings and niche relationships, their recommendations (such as developing natural resources, expanding industry, and strengthening infrastructure) appear to be generic suggestions without clear linkage to their empirical findings. The authors could establish a stronger logical connection by demonstrating how each recommendation specifically relates to their analytical results.

Response: We are very grateful for your valuable comments. We fully agree with the experts' opinions and have optimized the recommendations to ensure that each recommendation is closely centered around the empirical analysis results and is targeted to ensure that the logical connection between the analysis results and the recommendations is clearer and closer. (See lines 695-801)

Minor Issues,

1. Formula(2) need more clarification since it is somewhat different from the standard MacArthur model. (e.g., what do 'p', 'o', and 'R' represent)

Response: We appreciate your valuable comments. We have further clarified formula (2) based on your suggestions, supplemented the symbolic definitions, and explained its difference from the standard MacArthur model. (See lines 390-392)

2. The authors should report the specific β value used in their least squares method for combining subjective and objective weights.

Response: We are grateful for

---

## [Decision Letter · Decision Letter 1]

25 Feb 2025

PONE-D-24-42459R1Rural Sports Tourism Competitiveness in Shandong ProvincePLOS ONE

Dear Dr. Li,

Thank you for submitting your manuscript to PLOS ONE. After careful consideration, we feel that it has merit but does not fully meet PLOS ONE’s publication criteria as it currently stands. Therefore, we invite you to submit a revised version of the manuscript that addresses the points raised during the review process.

We look forward to receiving your revised manuscript.

Kind regards,

Yihao Li, Doctor

Academic Editor

PLOS ONE

Reviewers' comments:

Reviewer's Responses to Questions

**Comments to the Author**

1. If the authors have adequately addressed your comments raised in a previous round of review and you feel that this manuscript is now acceptable for publication, you may indicate that here to bypass the “Comments to the Author” section, enter your conflict of interest statement in the “Confidential to Editor” section, and submit your "Accept" recommendation.

Reviewer #3: All comments have been addressed

Reviewer #4: (No Response)

2. Is the manuscript technically sound, and do the data support the conclusions?

Reviewer #3: Yes

Reviewer #4: Partly

3. Has the statistical analysis been performed appropriately and rigorously? 

Reviewer #3: Yes

Reviewer #4: N/A

4. Have the authors made all data underlying the findings in their manuscript fully available?

Reviewer #3: Yes

Reviewer #4: Yes

5. Is the manuscript presented in an intelligible fashion and written in standard English?

Reviewer #3: Yes

Reviewer #4: Yes

6. Review Comments to the Author

Reviewer #3: The revised version shows significant improvement in clarity, logical flow, and overall readability.

While the majority of the previous concerns have been adequately addressed, there are still a few minor issues that require attention. I recommend a minor revision to address the following points:

1. Regarding the measurement of tourism resource quality (previous Major Issue #3):

To strengthen your theoretical foundation, it would be helpful to explore and cite examples of how resource diversity theory has been applied in related fields (such as sustainable tourism). These applications could provide valuable support for your methodology in tourism resource assessment.

2. Regarding the weight assignment (previous Major Issue #4):

While you have explained that the weights are based on resource influence, policy support, and reference to similar studies, it would be helpful to provide more specific justification for the 2-point difference between national and provincial levels or include references to similar weighting schemes in previous studies, if available

3. Regarding writing style:

There are several places throughout the manuscript where the language could be more concise and clear. For instance, the paragraph in lines 159-171 could be restructured to improve readability while maintaining the key theoretical framework and supporting references.

Reviewer #4: General comment –The authors have made commendable progress in revising their manuscript, notably enhancing the introduction’s depth, clarifying the theoretical foundation, and improving the methods section’s structure. These efforts have strengthened the justification for focusing on Shandong Province and better separated theoretical and methodological elements. However, to fully meet scholarly expectations and broaden the study’s impact, further refinements are needed. The title’s regional focus, methodological transparency, engagement with existing literature, and omission of limitations remain areas of concern. Addressing these through targeted revisions will elevate the manuscript’s quality and relevance to both regional and global audiences.

Comment 1 – The current title, "Rural Sports Tourism Competitiveness in Shandong Province," emphasizes a specific region, which may limit its perceived relevance to readers outside this context. To better reflect the study’s broader methodological and theoretical contributions, I recommend revising the title to something like "Evaluating Rural Sports Tourism Competitiveness: A Framework Applied to Shandong Province." This adjustment highlights the framework’s potential applicability elsewhere while retaining the case study focus, aligning with your textual claims of wider significance and attracting a more diverse readership.

Comment 2 – The "Determining subjective weights: AHP" subsection lacks detailed steps for the Analytic Hierarchy Process (AHP), stating only that they are context-dependent, which hinders transparency and replicability. I suggest including a concise summary of key AHP steps—such as hierarchy construction, pairwise comparisons, and consistency checks—along with a citation to a foundational source like Saaty (1980). If these details are already present elsewhere in the revised manuscript, ensure they are clearly integrated and accessible. Providing this clarity ensures readers can follow and replicate your subjective weighting process.

Comment 3 – While the discussion provides practical recommendations, it lacks comparisons with prior studies and a clear explanation of niche theory’s theoretical contributions to rural sports tourism competitiveness. I recommend integrating comparisons with works like Wang & Zhu (2019) or Jia et al. (2022) to show how your findings align or differ, and explicitly detailing how niche theory advances tourism competitiveness research. These additions will situate your study within the broader academic discourse and underscore its theoretical and empirical significance.

Comment 4 – The conclusion summarizes findings and strategies well but omits discussion of the study’s limitations and future research directions. I suggest adding a paragraph acknowledging limitations, such as the reliance on quantitative resource measures or potential subjective weighting biases, and proposing future steps, like testing the framework in other regions or incorporating qualitative data. Including these elements will enhance transparency, demonstrate academic integrity, and encourage further exploration of your work.

Comment 5 – The authors should avoid excessive dependence on AI tools, such as ChatGPT, when elaborating their manuscript. While AI can assist with drafting or structuring ideas, over-reliance may result in generic or shallow content that lacks the originality and critical depth required in academic writing. I encourage prioritizing human-led analysis and revisions to ensure the manuscript reflects your expertise and provides nuanced, authentic insights that meet scholarly standards.

7. PLOS authors have the option to publish the peer review history of their article (what does this mean? ). If published, this will include your full peer review and any attached files.

**Do you want your identity to be public for this peer review?** For information about this choice, including consent withdrawal, please see our Privacy Policy .

Reviewer #3: No

Reviewer #4: No

---

## [Author Response · Author response to Decision Letter 2]

28 Feb 2025

Response to Reviewers #3

I am most grateful to you for your valuable suggestions regarding this paper. I have used the highlighting function to indicate the revisions made to the manuscript.

1. Regarding the measurement of tourism resource quality (previous Major Issue #3):

To strengthen your theoretical foundation, it would be helpful to explore and cite examples of how resource diversity theory has been applied in related fields (such as sustainable tourism). These applications could provide valuable support for your methodology in tourism resource assessment.

Response: We are very grateful for your suggestions. Based on your suggestions, we have supplemented and cited relevant applications of resource diversity theory to provide a theoretical basis for the measurement of tourism resource quality. (Please see lines 178-182)

2. Regarding the weight assignment (previous Major Issue #4):

While you have explained that the weights are based on resource influence, policy support, and reference to similar studies, it would be helpful to provide more specific justification for the 2-point difference between national and provincial levels or include references to similar weighting schemes in previous studies, if available

Response: Thank you for your valuable comments. We agree with your point of view, and we have supplemented the relevant national standard documents, similar weighting schemes in previous studies, and similar research references in resource diversity theory to provide more specific reasons for the two differences between the national and provincial levels. (Please see lines 165-184)

3. Regarding writing style:

There are several places throughout the manuscript where the language could be more concise and clear. For instance, the paragraph in lines 159-171 could be restructured to improve readability while maintaining the key theoretical framework and supporting references.

Response: Thank you for your suggestion. Regarding the issue of language conciseness you raised, we have reorganized the relevant content you pointed out to ensure that while maintaining the integrity of the theoretical framework and citing key references, we reduce the problem of redundant sentences and improve the readability and clarity of the paragraphs. (Please see lines 148-164, 849-868)

I would like to express my gratitude once more for your invaluable contribution to the enhancement of this manuscript. Thank you!

Response to Reviewers #4

I am most grateful to you for your valuable suggestions regarding this paper. I have used the highlighting function to indicate the revisions made to the manuscript.

Comment 1

1.The current title, "Rural Sports Tourism Competitiveness in Shandong Province," emphasizes a specific region, which may limit its perceived relevance to readers outside this context. To better reflect the study’s broader methodological and theoretical contributions, I recommend revising the title to something like "Evaluating Rural Sports Tourism Competitiveness: A Framework Applied to Shandong Province." This adjustment highlights the framework’s potential applicability elsewhere while retaining the case study focus, aligning with your textual claims of wider significance and attracting a more diverse readership.

Response: Thank you for your valuable comments. Your suggestions are very pertinent. After comprehensive consideration by our team, we decided to change the title to your suggestion: "Evaluating Rural Sports Tourism Competitiveness: A Framework Applied to Shandong Province.". (Please see lines 1-3)

Comment 2

2. The "Determining subjective weights: AHP" subsection lacks detailed steps for the Analytic Hierarchy Process (AHP), stating only that they are context-dependent, which hinders transparency and replicability. I suggest including a concise summary of key AHP steps—such as hierarchy construction, pairwise comparisons, and consistency checks—along with a citation to a foundational source like Saaty (1980). If these details are already present elsewhere in the revised manuscript, ensure they are clearly integrated and accessible. Providing this clarity ensures readers can follow and replicate your subjective weighting process.

Response: Thank you for your valuable comments. Thank you for your valuable comments. We have optimized the detailed steps of AHP (Analytic Hierarchy Process), supplemented the detailed description of key steps, including hierarchy construction, pairwise comparisons, and consistency checks, and cited Saaty's classic literature to ensure the transparency and replicability of the method. At the same time, we have uploaded the relevant detailed data of the judgment matrix and consistency check to the project "Figure", named "Comparison Matrix" and "Consistency Check" respectively. (Please see lines 245-288, 1088-1092)

Comment 3

3. While the discussion provides practical recommendations, it lacks comparisons with prior studies and a clear explanation of niche theory’s theoretical contributions to rural sports tourism competitiveness. I recommend integrating comparisons with works like Wang & Zhu (2019) or Jia et al. (2022) to show how your findings align or differ, and explicitly detailing how niche theory advances tourism competitiveness research. These additions will situate your study within the broader academic discourse and underscore its theoretical and empirical significance.

Response: We are very grateful for your suggestions. We highly appreciate your suggestion, and we have added comparisons with studies such as Wang & Zhu (2019) or Jia et al. (2022) to present our research findings and detail the theoretical contribution of niche theory to tourism competitiveness and rural sports tourism competitiveness (please see lines 849-868)

Comment 4

4. The conclusion summarizes findings and strategies well but omits discussion of the study’s limitations and future research directions. I suggest adding a paragraph acknowledging limitations, such as the reliance on quantitative resource measures or potential subjective weighting biases, and proposing future steps, like testing the framework in other regions or incorporating qualitative data. Including these elements will enhance transparency, demonstrate academic integrity, and encourage further exploration of your work.

Response: Thank you for your valuable comments. We highly appreciate your suggestions. Based on the points you raised, we have made targeted supplements to the discussion of research limitations and future research directions. (Please see lines 869-878)

Comment 5

5. The authors should avoid excessive dependence on AI tools, such as ChatGPT, when elaborating their manuscript. While AI can assist with drafting or structuring ideas, over-reliance may result in generic or shallow content that lacks the originality and critical depth required in academic writing. I encourage prioritizing human-led analysis and revisions to ensure the manuscript reflects your expertise and provides nuanced, authentic insights that meet scholarly standards.

Response: Thank you for your valuable comments. We completely agree with your point about avoiding over-reliance on AI tools and understand the importance of originality and critical depth in academic writing. During the writing process, we did use AI tools to assist in organizing ideas, but we always adhere to human-led analysis and revision, and all content has been deeply analyzed and revised multiple times by our team to ensure that it meets academic standards and reflects our expertise.

I would like to express my gratitude once more for your invaluable contribution to the enhancement of this manuscript. Thank you!

---

## [Decision Letter · Decision Letter 2]

2 Apr 2025

PONE-D-24-42459R2Evaluating Rural Sports Tourism Competitiveness: A Framework Applied to Shandong ProvincePLOS ONE

Dear Dr. Li,

Thank you for submitting your manuscript to PLOS ONE. After careful consideration, we feel that it has merit but does not fully meet PLOS ONE’s publication criteria as it currently stands. Therefore, we invite you to submit a revised version of the manuscript that addresses the points raised during the review process.

 Please submit your revised manuscript by May 17 2025 11:59PM. If you will need more time than this to complete your revisions, please reply to this message or contact the journal office at plosone@plos.org . Please include the following items when submitting your revised manuscript:

We look forward to receiving your revised manuscript.

Kind regards,

Yihao Li, Doctor

Academic Editor

PLOS ONE

Reviewers' comments:

Reviewer's Responses to Questions

**Comments to the Author**

1. If the authors have adequately addressed your comments raised in a previous round of review and you feel that this manuscript is now acceptable for publication, you may indicate that here to bypass the “Comments to the Author” section, enter your conflict of interest statement in the “Confidential to Editor” section, and submit your "Accept" recommendation.

Reviewer #3: All comments have been addressed

Reviewer #4: (No Response)

2. Is the manuscript technically sound, and do the data support the conclusions?

Reviewer #3: Yes

Reviewer #4: Partly

3. Has the statistical analysis been performed appropriately and rigorously? 

Reviewer #3: Yes

Reviewer #4: N/A

4. Have the authors made all data underlying the findings in their manuscript fully available?

Reviewer #3: Yes

Reviewer #4: Yes

5. Is the manuscript presented in an intelligible fashion and written in standard English?

Reviewer #3: Yes

Reviewer #4: Yes

6. Review Comments to the Author

Reviewer #3: The authors' adoption of Niche Theory to analyze the rural sports industry provides a novelperspective. Throughout the two rounds of revisions, the authors have effectively addressed ambiguities, providing additional explanations and incorporating relevant literature to strengthen their arguments. While there are still some language issues, the paper offers practical guidance for addressing real-world issues in rural sports development.

Reviewer #4: General comment –The authors have made commendable progress in addressing comments from the previous round of revisions; however, several key issues remain that require further attention. Specifically, although comparisons with prior studies (Wang & Zhu, 2019; Jia et al., 2022) have been introduced, the discussion section still lacks sufficient citations and explicit theoretical contrasts with relevant literature. It is recommended that the authors reorganise the discussion to clearly summarise key findings, explicitly contrast them with existing literature, and highlight theoretical contributions, particularly regarding niche theory. Additionally, the manuscript still lacks dedicated sections for theoretical and practical implications, as well as explicit acknowledgment of study limitations and future research directions. Addressing these aspects clearly and systematically will further enhance the manuscript’s scholarly rigour and significance.

Comment 1 – The authors have indeed made notable progress by incorporating comparisons with prior studies like Wang & Zhu (2019) and Jia et al. (2022), as suggested. However, the discussion still predominantly lacks sufficient citations and detailed contrasts with relevant literature, which prevents readers from fully understanding how your findings confirm, differ from, or extend prior research.

To further enhance the academic rigour of your discussion section, I recommend clearly structuring it into three key components: (1) summarising your key findings succinctly, (2) explicitly comparing and contrasting your findings with those from existing studies, citing precise similarities or divergences, and (3) explaining the theoretical implications of these comparisons, particularly in relation to niche theory. Each of these components should be addressed separately and distinctly, accompanied by clear references to relevant literature. Increasing the depth of engagement with prior research will strengthen your manuscript’s theoretical contributions and help clarify the unique aspects your study brings to rural sports tourism competitiveness research.

For example, you could explicitly state how your conclusions about resource endowment align with or deviate from Wang & Zhu (2019), and whether your findings regarding niche overlap and municipal competitiveness offer new insights compared to Jia et al. (2022). Furthermore, explicitly identifying how your study fills existing theoretical gaps or addresses unanswered questions in niche theory literature would significantly strengthen your manuscript.

By following this approach, you will better highlight both the originality and theoretical contribution of your research.

Comment 2 –While the manuscript now appropriately discusses the results and limitations, it lacks a clearly structured subsection dedicated explicitly to Theoretical and Practical Implications. To enhance the paper's clarity and utility, I strongly recommend adding a separate subsection, clearly outlining:

Theoretical Implications: Specify how this research advances niche theory or contributes to theoretical understanding of rural sports tourism competitiveness. Clearly articulate the novel theoretical insights or extensions beyond existing studies, particularly highlighting how niche theory enriches the broader tourism competitiveness discourse.

Practical Implications: Provide concrete, actionable recommendations aimed at policymakers, tourism planners, and practitioners, clarifying exactly how the findings can inform policy decisions, marketing strategies, or infrastructure investments.

Presenting these implications explicitly in a dedicated subsection will significantly improve the manuscript’s value for both researchers and practitioners.

Comment 3 –Although the authors briefly discussed limitations and future research directions within the "Discussion" section, the manuscript lacks a dedicated "Limitations and Future Research" subsection. This section is essential for explicitly acknowledging the boundaries of your research and guiding readers toward valuable next steps. To enhance clarity and rigor, I strongly recommend creating a separate subsection titled "Limitations and Future Research," clearly outlining limitations related to your data sources, methodological constraints, potential biases, and generalisability of your findings. Additionally, future research directions should be explicitly articulated, including suggestions on expanding geographic scope, integrating qualitative methods, or exploring other dimensions not fully addressed in your study. This dedicated subsection would significantly enhance the clarity and completeness of the manuscript, assisting readers in identifying research gaps and opportunities for further exploration.

7. PLOS authors have the option to publish the peer review history of their article (what does this mean? ). If published, this will include your full peer review and any attached files.

**Do you want your identity to be public for this peer review?** For information about this choice, including consent withdrawal, please see our Privacy Policy .

Reviewer #3: No

Reviewer #4: No

---

## [Author Response · Author response to Decision Letter 3]

4 Apr 2025

Reviewer #3

I am most grateful to you for your valuable suggestions regarding this paper. I have used the highlighting function to indicate the revisions made to the manuscript.

1.The authors' adoption of Niche Theory to analyze the rural sports industry provides a novel perspective. Throughout the two rounds of revisions, the authors have effectively addressed ambiguities, providing additional explanations and incorporating relevant literature to strengthen their arguments. While there are still some language issues, the paper offers practical guidance for addressing real-world issues in rural sports development.

Response: We are very grateful for your suggestions, which are of great significance to the improvement of our paper. In combination with your suggestions, we have systematically sorted out the language issues. (Please see lines 682-934)

I would like to express my gratitude once more for your invaluable contribution to the enhancement of this manuscript. Thank you!

Reviewer #4

I am most grateful to you for your valuable suggestions regarding this paper. I have used the highlighting function to indicate the revisions made to the manuscript.

Comment 1

1.The authors have indeed made notable progress by incorporating comparisons with prior studies like Wang & Zhu (2019) and Jia et al. (2022), as suggested. However, the discussion still predominantly lacks sufficient citations and detailed contrasts with relevant literature, which prevents readers from fully understanding how your findings confirm, differ from, or extend prior research.

To further enhance the academic rigour of your discussion section, I recommend clearly structuring it into three key components: (1) summarising your key findings succinctly, (2) explicitly comparing and contrasting your findings with those from existing studies, citing precise similarities or divergences, and (3) explaining the theoretical implications of these comparisons, particularly in relation to niche theory. Each of these components should be addressed separately and distinctly, accompanied by clear references to relevant literature. Increasing the depth of engagement with prior research will strengthen your manuscript’s theoretical contributions and help clarify the unique aspects your study brings to rural sports tourism competitiveness research.

For example, you could explicitly state how your conclusions about resource endowment align with or deviate from Wang & Zhu (2019), and whether your findings regarding niche overlap and municipal competitiveness offer new insights compared to Jia et al. (2022). Furthermore, explicitly identifying how your study fills existing theoretical gaps or addresses unanswered questions in niche theory literature would significantly strengthen your manuscript.

By following this approach, you will better highlight both the originality and theoretical contribution of your research.

Response: Thank you for your valuable comments. We highly appreciate your suggestions. We have restructured 5 according to your suggestions on the discussion section and elaborated on the key parts you mentioned in 5.1 and 5.2.1. (Please see lines 683-703, 705-737)

Comment 2

2. While the manuscript now appropriately discusses the results and limitations, it lacks a clearly structured subsection dedicated explicitly to Theoretical and Practical Implications. To enhance the paper's clarity and utility, I strongly recommend adding a separate subsection, clearly outlining:

Theoretical Implications: Specify how this research advances niche theory or contributes to theoretical understanding of rural sports tourism competitiveness. Clearly articulate the novel theoretical insights or extensions beyond existing studies, particularly highlighting how niche theory enriches the broader tourism competitiveness discourse.

Practical Implications: Provide concrete, actionable recommendations aimed at policymakers, tourism planners, and practitioners, clarifying exactly how the findings can inform policy decisions, marketing strategies, or infrastructure investments.

Presenting these implications explicitly in a dedicated subsection will significantly improve the manuscript’s value for both researchers and practitioners.

Response: Thank you for your valuable comments. We highly appreciate your suggestions. Based on your suggestions, we have set the theoretical implications and practical significance you mentioned to 5.2.2, 5.2.3 and 5.3 respectively to clearly explain the questions you raised. (Please see lines 738-751, 752-768, 769-875)

Comment 3

3.Although the authors briefly discussed limitations and future research directions within the "Discussion" section, the manuscript lacks a dedicated "Limitations and Future Research" subsection. This section is essential for explicitly acknowledging the boundaries of your research and guiding readers toward valuable next steps. To enhance clarity and rigor, I strongly recommend creating a separate subsection titled "Limitations and Future Research," clearly outlining limitations related to your data sources, methodological constraints, potential biases, and generalisability of your findings. Additionally, future research directions should be explicitly articulated, including suggestions on expanding geographic scope, integrating qualitative methods, or exploring other dimensions not fully addressed in your study. This dedicated subsection would significantly enhance the clarity and completeness of the manuscript, assisting readers in identifying research gaps and opportunities for further exploration.

Response: Thank you for your valuable comments. We highly appreciate your suggestions. Based on your suggestions, we set "Limitations and Future Research" to 6 separately, and re-summarized and sorted out the limitations and future research respectively. (Please see lines 909-934)

I would like to express my gratitude once more for your invaluable contribution to the enhancement of this manuscript. Thank you!

---

## [Decision Letter · Decision Letter 3]

16 Apr 2025

Evaluating Rural Sports Tourism Competitiveness: A Framework Applied to Shandong Province

PONE-D-24-42459R3

Dear Dr. Li,

We’re pleased to inform you that your manuscript has been judged scientifically suitable for publication and will be formally accepted for publication once it meets all outstanding technical requirements.

Kind regards,

Yihao Li, Doctor

Academic Editor

PLOS ONE

Additional Editor Comments (optional):

Reviewers' comments:

Reviewer's Responses to Questions

**Comments to the Author**

1. If the authors have adequately addressed your comments raised in a previous round of review and you feel that this manuscript is now acceptable for publication, you may indicate that here to bypass the “Comments to the Author” section, enter your conflict of interest statement in the “Confidential to Editor” section, and submit your "Accept" recommendation.

Reviewer #4: All comments have been addressed

2. Is the manuscript technically sound, and do the data support the conclusions?

Reviewer #4: Yes

3. Has the statistical analysis been performed appropriately and rigorously? 

Reviewer #4: Yes

4. Have the authors made all data underlying the findings in their manuscript fully available?

Reviewer #4: Yes

5. Is the manuscript presented in an intelligible fashion and written in standard English?

Reviewer #4: Yes

6. Review Comments to the Author

Reviewer #4: I would like to sincerely commend the authors for the thoughtful and thorough revisions made to the manuscript “Evaluating Rural Sports Tourism Competitiveness: A Framework Applied to Shandong Province.” It is clear that the authors have carefully considered the previous feedback and have significantly enhanced the manuscript’s clarity, coherence, and academic contribution.

The restructured discussion section is notably improved, now providing a more systematic comparison with prior studies and a clearer articulation of how the findings extend existing knowledge. The theoretical implications are now well integrated, and the application of niche theory is both innovative and convincingly justified. The authors have also strengthened the practical relevance of the work by offering concrete recommendations tailored to policymakers, planners, and practitioners.

The addition of clearly defined subsections on theoretical contributions, practical implications, and limitations and future research has brought greater structure to the manuscript and will be particularly helpful to readers seeking to understand both the scope and significance of the study. These enhancements contribute to a more impactful and publishable piece of research.

Overall, the current version reflects a well-developed and original contribution to the literature on rural tourism competitiveness, and I am pleased with the progress made in this round of revisions.

7. PLOS authors have the option to publish the peer review history of their article (what does this mean? ). If published, this will include your full peer review and any attached files.

**Do you want your identity to be public for this peer review?** For information about this choice, including consent withdrawal, please see our Privacy Policy .

Reviewer #4: No

---

## [Editor Report · Acceptance letter]

PONE-D-24-42459R3

PLOS ONE

Dear Dr. Li,

I'm pleased to inform you that your manuscript has been deemed suitable for publication in PLOS ONE. Congratulations! Your manuscript is now being handed over to our production team.

Kind regards,

on behalf of

Dr. Yihao Li

Academic Editor

PLOS ONE